# Review Article: A Comparison of Flood and Earthquake Vulnerability Assessment Indicators

Marleen C. de Ruiter[1], Philip J. Ward[1], James E. Daniell[2], Jeroen C. J. H. Aerts[1]

[1]Institute for Environmental Studies (IVM), Vrije Universiteit Amsterdam, Amsterdam, 1081HV, The Netherlands
[2]Geophysical Institute and Center for Disaster Management and Risk Reduction Technology, Karlsruhe Institute of Technology (KIT), Karlsruhe, 76344, Germany

*Correspondence to:* Marleen C. de Ruiter (m.c.de.ruiter@vu.nl)

**Abstract.** In a cross-disciplinary study, we carried out an extensive literature review to increase understanding of vulnerability indicators used in the disciplines of earthquake- and flood vulnerability assessments. We provide insights into potential improvements in both fields by identifying and comparing quantitative vulnerability indicators grouped into physical- and social categories. Next, a selection of index- and curve-based vulnerability models that use these indicators are described, comparing several characteristics such as temporal- and spatial 15 aspects. Earthquake vulnerability methods traditionally have a strong focus on object-based physical attributes used in vulnerability curve-based models, while flood vulnerability studies focus more on indicators applied to aggregated land-use classes in curve-based models. In assessing the differences and similarities between indicators used in earthquake and flood vulnerability models, we only include models that separately assess either of the two hazard types. Flood vulnerability studies could be improved using approaches from earthquake studies, 20 such as developing object-based physical vulnerability curve assessments and incorporating time-of-the-day based building occupation patterns. Likewise, earthquake assessments could learn from flood studies by refining their selection of social vulnerability indicators. Based on the lessons obtained in this study, we recommend future studies for exploring risk assessment methodologies across different hazard types.

## 1 Introduction

Recent decades have seen a sharp global increase in the economic risk associated with floods and earthquakes, although it should be noted that both earthquake and flood related fatalities might be decreasing. UNISDR (2009) defines this risk as: "the probability of harmful consequences, or expected losses (deaths, injuries, property, livelihoods, economic activity disrupted or environment damaged) resulting from interactions between natural or human-induced hazards and vulnerable conditions". Based on previous work by Crichton (1999) and Kron (2005), 30 this risk has been formalised in many studies and frameworks (e.g. UNISDR, 2009; Mechler and Bouwer, 2014) using the following Eq. (1):

$$Risk = f(Hazard, Exposure, Vulnerability) \,, \tag{1}$$

where *hazard* is defined as 'A potentially damaging physical event, phenomenon or human activity that may cause the loss of life or injury, property damage, social and economic disruption or environmental degradation';

*Exposure* is defined as 'People, property, systems, or other elements present in hazard zones that are thereby subject to potential losses'; and *Vulnerability* as the set of conditions and processes resulting from physical, social, economic, and environmental factors, which increase the susceptibility of a community '(people and assets) to the impact of hazards' (UNISDR 2009). Both in the domain of flooding and earthquakes, improving methods to assess vulnerability is seen as the 'missing link' for increasing our understanding of risk (Douglas,

2007; Jongman et al., 2015). A recent review of the Sendai framework by Mysiak et al. (2016) shows that one of the key components required, is to identify and increase understanding of the main vulnerability indicators that drive risk.

In this paper, we use the widely applied definition of vulnerability as provided by UNISDR (2009). The paper specifically does not aim to produce another definition of vulnerability and we gratefully acknowledge the broad

literature on vulnerability and previous discussions of definitions and conceptualizations of vulnerability (e.g. Alexander 1997; Cardona 2004; Cutter et al., 2003; Adger, 2006; Barroca et al., 2006; Birkmann et al., 2007; Hinkel, 2011).

Many studies have suggested that the observed increase in risk in recent decades is mainly due to the increase in

exposure of assets and people in hazard prone areas, and an increase in wealth (Pielke Jr and Downton, 2000; Kron, 2005; UNISDR, 2011; IPCC, 2012; Doocy et al., 2013b; Blaikie et al., 2014; MunichRe, 2014; Visser et al., 2014; GFDRR, 2016). To date, most studies on flood risk have found little signal for increasing hazard in the last decades (e.g. Kundzewicz et al., 2014; Jongman et al., 2015). However, recent research suggests that this could be due to the fact these studies have not accounted for changes in vulnerability over time (e.g. Mechler and Bouwer,

2014; Jongman et al., 2015) and the impact of risk reduction policies on flood damage and societal flood vulnerability is not well understood (Pielke Jr and Downton, 2000). Indeed, the quantification of vulnerability in risk assessments is known to be extremely difficult, which is why most studies assume constant vulnerability over time.

There are two distinct paradigms in assessing vulnerability: the natural sciences and the social sciences (Roberts et al., 2009). The former considers the human system to be passive, while exposed elements have varying vulnerability to a hazard which can differ in magnitude and is considered to be an active agent. In the social sciences approach to assessing vulnerability, the focus is on the coping capacity and resilience of the human system (Roberts et al., 2009). While acknowledging the studies that further subdivide vulnerability into resilience

and susceptibility, or that consider resilience to be vulnerability's counterpart (e.g. Fuchs 2009), we asses vulnerability as it is defined by UNISDR (2009), but we do account for both physical and socio-economic indicators of vulnerability.

When focusing on the quantification of vulnerability to (fluvial) flooding, as part of a flood risk model, there are two main approaches: (a) vulnerability indices; and (b) vulnerability curves (Messner et al., 2007; Kannami, 2008; Merz et al., 2010; Nasiri et al., 2013). Although the field of vulnerability assessment is wider (Adger 2006; Birkmann 2007), we here focus on these two main types of quantitative vulnerability assessment methods that are commonly used in risk assessment models. Both approaches use one or more indicators that influence vulnerability and are used as measures of vulnerability (Cutter et al., 2003). Well-known contributions to index-based vulnerability assessments (not necessarily focusing on one hazard type) have been made by Cutter (2003), Davidsson and Shah (1997), Coburn and Spence (1994 and 2002), and many others. Vulnerability indices are sometimes combined with statistical multi-variate methods to find correlations between empirical losses from natural hazards (e.g. Carreño et al., 2007). In flood risk modeling, there are numerous studies that have assessed the influence of temporal and spatial changes in hazard and exposure on risk, using risk models or risk-based indicators (e.g. Apel et al., 2004; Bouwer et al., 2007; Bouwer, 2011; IPCC, 2012; De Moel et al., 2015; Jongman et al., 2015). Most of the risk models, however, make simple assumptions on quantifying vulnerability, and have largely refrained from considering (changing) vulnerability as a potential cause of the growing impacts of floods (Koks et al., 2015b; Mechler and Bouwer, 2014). Several key challenges with the quantification of vulnerability to flooding include: (1) difficulties in developing meaningful and quantifiable indicators of vulnerability; (2) a lack of available and accurate data to measure those indicators, and the fact that the required data are often only available at highly aggregated levels; and (3) a lack of empirical data on flood losses to relate losses (damage) to vulnerability (Birkmann 2006; Thieken et al., 2008; Notaro et al., 2014).

Compared to other natural hazards, the quantification of vulnerability is most detailed for earthquake risk assessment models although challenges remain (Douglas 2007; Roberts et al., 2009). Historically, the assessment of physical vulnerability (often referred to as 'fragility') is well-developed and recently attempts have also been made to improve the quantification of social vulnerability (Sauter and Shah, 1987; Tiedemann, 1991; Yücemen et al., 2004; Carreño et al., 2005; Douglas, 2007; Roberts et al., 2009). As with flood risk assessment, most of the methods to assess earthquake vulnerability are either based on indices or vulnerability curves. Earthquake vulnerability assessments traditionally have a very strong focus on the physical vulnerability of individual buildings, their construction, and specific structural characteristics. Examples include: the number of stories; their ability to resist seismic lateral forces as a primary cause of building damage; and casualties caused by building collapse (Coburn and Spence, 2002). Damage to buildings is generally the sole indicator used to predict economic and social losses (Kircher et al., 2006).

The main goal of this study is to conduct a literature review to provide insights into how vulnerability indicators (both physical and social) are used in quantitative flood- and earthquake risk assessment models by comparing two different methods for quantitatively assessing vulnerability in flood and earthquake risk assessment models (i.e. curve- and index-based vulnerability assessments). It therefore does not aim to provide a comprehensive overview of all vulnerability indicators in the domain of floods or earthquakes. Instead, we analyze only those

indicators that have been addressed in both modeling domains and systematically assess the differences in using those indicators in both flood vulnerability and earthquake risk models. We recognize that the study of cascading events is an important, emerging field as discussed extensively in Pescaroli and Alexander (2016), however our focus is on single events only. More specifically, we analyze which vulnerability indicators have been addressed in such quantitative methods by comparing the fields of flood and earthquake risk assessment. Through this comparison, we hope that both fields can learn from each other's respective approaches, further developing vulnerability as an important component in risk modeling.

The remainder of this paper is organized as follows: Sect. 2 describes the methods followed to compare the different vulnerability assessment methods, including a discussion of several well-known earthquake and flood risk or vulnerability assessment methods. In Sect. 3, we discuss main differences and similarities between earthquake and flood vulnerability indicators. Finally, a brief conclusion and recommendations section follows.

## 2    Identifying different vulnerability indicators and models for comparison

In this section, we describe the methods that we have used to structure an extensive literature review to compare vulnerability assessment models in both flood risk- and earthquake assessments. In Sect. 2.1, we provide an overview of the main vulnerability indicators (categorized as physical or social) that have been used to quantify flood and earthquake vulnerability. Next, in Sect. 2.2, we describe the two modeling types that use these indicators to quantify vulnerability: vulnerability curve models and index-based vulnerability models.

### 2.1    Vulnerability Indicators

Several studies have discussed the approach to, and potential pitfalls in, defining different indicator categories (e.g. Davidsson and Shah, 1997; Bruneau et al., 2003; Birkmann, 2007). Bruneau et al. (2003) suggest a framework for the quantitative assessment of seismic resilience consisting of the following four interrelated dimensions of community resilience for which there exist no single measure (note: their definition of resilience overlaps in part with the definition of vulnerability used in this paper): technical, organization, social, and economic. Davidsson and Shah (1997) acknowledge the necessity of the development of "an index of vulnerability". Their Earthquake Disaster Risk Index (EDRI), a composite index, allows for the inclusion of different factors of vulnerability (i.e. physical infrastructure, population, economy and social-political system) (Davidsson and Shah, 1997). Davidsson and Shah (1997) too, acknowledge that factors (or classes) of vulnerability are not distinct entities and that there are many interactions, overlaps and contradictions between indicators from the different classes.  While acknowledging the difficulties in categorizing  vulnerability, we classify vulnerability indicators, similar to many flood and earthquake vulnerability assessments, in two main classes: (a) physical indicators that pertain directly to characteristics of the exposed assets, namely infrastructure and lifelines (including transportation infrastructure, utility lifelines, and essential lifelines) and buildings (including structural elements, occupancy, and environment related factors); and (b) social indicators, which

include here: demographics, awareness, socio-economics, and institutional factors (e.g. Mileti, 1999; Cutter et al., 2003; Adger, 2006; Messner and Meyer, 2006; Roberts et al., 2009; Balica et al., 2012).

Vulnerability indicators can be categorized in direct versus indirect indicators. Where the engineering community
has mainly addressed direct (or physical) damage, the economic research community has mainly addressed indirect (economic) damages (Koks et al., 2015a).  In recent years, it has become more common for damage models to integrate both approaches (Koks et al., 2015a). Koks et al. (2015a) explain that many studies have been developed that assess the direct consequences of flooding but to a much lesser extent incorporate the indirect consequences of flooding. Direct indicators of vulnerability are related to the immediate physical contact of a
flooding to humans, properties and the natural environment (Messner and Meyer, 2006; Hiete and Merz, 2009). Indirect indicators on the other hand focus on the consequential effects of direct damage, often focussing on production losses due to economic interruptions in and outside the disaster-struck area (Hiete and Merz, 2009; Koks et al., 2015a). However, indirect vulnerability indicators are often omitted from flood vulnerability studies due to the lack of available empirical data (Penning-Rosswel et al., 2003). Heite and Merz (2009) developed a
conceptual framework to assess indirect vulnerability indicators for industrial sectors. The use of an indicator-based approach makes it possible to account for indirect components of vulnerability (Khazai et al., 2013). Therefore, indirect indicators have been included in our study but occur much less frequently. Furthermore, direct and indirect vulnerability, can each be subdivided into tangible and intangible indicators. Tangible indicators can be expressed in monetary values whereas intangible indicators are non-monetary (Messner and Meyer, 2006).
Unlike tangible flood effects, flood vulnerability assessments incorporate intangible indicators to a much lesser extent as it often requires a monetization of indicators such the value of human life, health or environmental aspects (Messner and Meyer, 2006). Adger (1999) discusses how some indicators of vulnerability can also be both direct and indirect, such as social inequality, which can be a direct measure of the coping capacity of a household or community to respond to a disaster but it can also be interpreted as an indirect measure of increased poverty
and insecurity. Therefore, we have decided to omit the classification of indicators between direct and indirect as well as tangible versus intangible from this paper.

### 2.1.1    Physical Vulnerability

The physical factor of vulnerability is the most thoroughly researched segment of vulnerability science, in part
because physical vulnerability is more easily quantifiable than social vulnerability (Notaro et al., 2014), and relates to the physical vulnerability of the assets exposed to natural hazards – in our case floods and earthquakes. In accordance with several of the studies reviewed, we make a distinction in three main exposed assets: (a) infrastructure and lifelines; (b) buildings and their structural and occupancy components; and (c) environment (e.g. Davidson and Shah, 1997; Mileti 1999; Carreño et al., 2007; Douglas 2007).

**Infrastructure and lifelines indicators**

In terms of infrastructure assets, we further specify *Transportation infrastructure* (e.g. highways, railways, ports), *Utility lifelines* (e.g. potable water, waste water, electric power, oil systems), and *Essential facilities* (e.g. hospitals, police and fire stations, and schools) (FEMA 2013a and 2013b). Quantifiable physical vulnerability indicators for infrastructure for both earthquakes and floods include: (a) structural indicators, such as the length of railways and public roads in operation; and location indicators, such as accessibility of facilities or the closeness of utilities to another utility (e.g. Rashed and Weeks, 2003; Peng, 2012). As mentioned, there are challenges in grouping indicators in distinct categories. Some studies perceive lifeline vulnerability as part of social vulnerability (e.g. Cutter et al., 2003; Holand 2014). For example, Holand (2014) defines lifeline vulnerability as the aspects of social vulnerability that are influenced by lifeline failure and he reviews common indicators used. He argues that there has been little discussion on how to measure lifeline vulnerability and distinguishes three lifeline indicator categories: (1) indicators addressing lifeline density and financial impacts caused by a natural disaster; (2) indicators measuring network redundancy and the potential for losing connectivity; and (3) indicators measuring travel time to facilities that provide critical services. Many of the studies reviewed by Holand (2014) group lifeline indicators with built environment or other physical indexes.

**Building structural and occupancy indicators**

The vulnerability of buildings can be described using two indicator groups: *structural elements* and *occupancy indicators*. Structural elements comprise of, for example, building type, material, age, and number of floors (e.g. Giovinazzi and Lagomarsino, 2004; Kircher et al., 2006; Porter et al., 2008; Duzgun et al., 2011). Building occupancy refers to the building-usage type , for example, commercial, industry or residential. These occupancy types determine the potential values of the losses from a hazard (e.g. Kircher et al., 2006; FEMA 2013a and 2013b).

**Environmental indicators**

The vulnerability of both infrastructure and buildings is influenced by their environmental characteristics. For example, the proximity of a building to a potential contaminating site may affect vulnerability (e.g. Colombi et al., 2008; Damm 2009), exemplified by the Elbe floods of 2002, when relatively minor damage (i.e. the damage as percentage of the total damage) was caused due to oil tanks that were buried in gardens of houses, but that were floating and leaking due to flood waters (Kreibich et al., 2005; Müller and Thieken, 2005).

**2.1.2    Social Vulnerability**

The definition of social vulnerability is much debated (Birkmann 2007).  Hinkel (2011) states that although the debate around the conceptualization of social vulnerability continues to exist, agreement seems to have been reached on social vulnerability being context-specific and place-based as defined by Cutter et al. (2003). In this paper, we therefore use the definition of social vulnerability as provided by Cutter et al. (2003), where social vulnerability consists of social inequalities (i.e. social factors that influence peoples' susceptibility) and place inequality (i.e. factors such as urbanization and economic vitality that impact the social vulnerability of a place).

Tate (2012) argues that in more recent years, there has been an increase in studies aiming to develop social vulnerability indices to quantify the social dimensions of natural hazard vulnerability. Nonetheless, social

vulnerability is studied to a lesser extent than the physical vulnerability factors due to the lack of empirical data available to quantify social vulnerability, especially at the more detailed household levels (e.g. Cutter et al., 2003). As a result, social vulnerability is often expressed at more aggregated levels, using vulnerability indicators such as age, ethnicity and welfare levels of communities and countries (Cutter et al., 2003; Blaikie et al., 2014; de Sherbinin and Bardy, 2015). Two research communities have assessed social vulnerability quite extensively: the

climate change adaptation (CCA) community and the disaster risk reduction (DRR) research community (Turner et al., 2003; Thomalla et al., 2006; Mercer, 2010; Dewan, 2013). Concepts from both communities have become increasingly intertwined, integrating concepts of resilience and adaptive- or coping-capacity (e.g. Turner et al., 2003; Deressa, Hassan and Ringler, 2008; Kienberger et al., 2009; Merz et al., 2010; Scheuer et al., 2011; Brink and Davidson, 2015). Birkmann et al., (2013) provide an extensive overview of vulnerability perspectives and

discuss the framing of vulnerability by both the DRR and CCA communities. Since many risk assessment models use the concept of susceptibility in assessing vulnerability (Birkmann et al., 2013) and since this is in line with the UNISDR (2009) definition of vulnerability, we will exclude a focus on resilience as a separate concept.

Reviewing the existing studies, there is no consensus on which aspects to include in social vulnerability. Many

studies incorporate different combinations of social indicators (such as vulnerable age groups, population density and population growth) with political, environmental and/or economic indicators (e.g. Davidsson and Shah, 1999; Cardona 2006; Peduzzi et al., 2009). Based on this, we here distinguish four main social vulnerability indicator groups: demographic, awareness and preparedness, socio-economic, and institutional and political vulnerability. However, as mentioned before, we recognize that indicator categories are not clear cut and overlaps continue to

exist (Davidsson and Shah, 1997).

**Demographic indicators**

Demographic indicators refer to the size, structure, and distribution of populations, and related spatial or temporal changes in them in response to natural hazards. For example, for determining social vulnerability to earthquakes,

the 'vulnerable age' indicator is often used (e.g. Davidson and Shah, 1997; Schmidtlein et al., 2011).

**Awareness indicators**

Research has shown that risk perception is an important factor for households to determine their level of preparation for natural hazard events (e.g. Balica et al., 2012; Bubeck et al., 2012). For example, the experience

with previous events has a positive effect on the awareness level (Balica et al., 2009). In addition, access to information sources, such as TV, determines the knowledge and awareness of the hazard (e.g. Balica et al., 2009; Brink and Davidson, 2015). Education level was found to not only influence peoples' socio-economic vulnerability (e.g. Cutter et al., 2003) but also household awareness and preparedness levels (Rüstemli and Karanci, 1999; Shaw et al., 2004).


**Socio-Economic indicators**

Societal and individual wealth are important indicators in determining peoples' social vulnerability to natural disasters (de Sherbinin and Bardy, 2015). For example, research shows that relatively high-income households have a higher demand for hazard insurance, or more often implement damage mitigation measures (Botzen and

Van den Bergh, 2012; Bubeck et al., 2012) as they tend to be more exposed to coastal flooding (de Sherbinin and Bardy, 2015). In their SREX report, the IPCC (2012) recognizes the economic dimensions of vulnerability as being separate from the social dimensions while recognizing the strong correlation between human vulnerability and economic indicators such as poverty. Other studies lump social, economic and environmental indicators of vulnerability together often referring to them as socio-economic (e.g. Peduzzi et al., 2009) or include economic

indicators such as GDP in the broader concept of social vulnerability (e.g. Hinkel 2011). Therefore, we refer to this category as socio-economic and include the indicators into the overall category of social vulnerability. Examples of indicators in this category are GDP, income, or percentage of unemployed people (e.g. Davidson and Shah, 1997; Peduzzi et al., 2009; Hinkel 2011; Peng, 2012).

**Institutional and political indicators**

Indicators that refer to institutional and political factors are related to a certain level of planning and preparing for natural hazards. For example, strong (spatial-) planning regulations may be an indicator that building codes and zoning protocols have been developed and enforced (e.g. Cutter et al., 2003; Blaikie et al., 2014).

### 2.2 Vulnerability models

This section discusses a selection of earthquake and flood vulnerability assessment models. Hollenstein (2005) reviewed vulnerability models for a wide range of natural hazards and found that there were far more earthquake vulnerability models (100+) than flood models (less than 20). We have aimed to include an equal number of earthquake and flood vulnerability models. Vulnerability models use indicators from Sect. 2.1, combining information on the hazard, exposure and vulnerability indicators (e.g. Carreño et al., 2007). We use the

categorization of vulnerability methods as recognized in the literature (Messner et al., 2007; Merz et al., 2010; Nasiri et al., 2013), which distinguishes two main vulnerability modeling types: index-based models and models that use vulnerability curves. It should be noted however, that in some studies an index is generated and subsequently incorporated in a vulnerability curve (e.g. Giovinazzi and Lagomarsino, 2004). In those cases, we classified the indicator used to construct the index in the index-based models category. For a detailed review of

Earthquake Loss Estimation software packages, we refer to Daniell (2011) and for an overview of flood damage models to Jongman et al. (2012). For this study, we only include risk assessment models that focus on either one of the two hazard types, or models that consist of two separate segments for each hazard type, to focus on assessing the differences or similarities in indicator usage between flood and earthquakes. We here provide the main characteristics of such models, and describe a few important ones in more detail.

### 2.2.1    Index-based vulnerability models

This category includes models that assess vulnerability based on statistical data for the indicators listed in Sect. 2.1. These models sum different vulnerability indicators into one composite index, which then shows the vulnerability of a household, community, or country to natural hazards (Birkmann, 2007). These indicators are often used in statistical analyses to find relations between the vulnerability index and empirical losses. Simple examples are statistical analyses between damages (or fatalities) and a second variable such as *the number of buildings in need of large repair* in an area as used in the Global Earthquake Model (GEM) (e.g. Burton and Silva, 2014; Silva et al., 2014a and 2014b). Below we give several examples:

- The Flood Vulnerability Index (FVI) was developed by Connor and Hiroki (2005) and adapted in subsequent studies (Balica et al., 2009; 2010 and 2012). The FVI combines different cause and effect factors and consists of four components (i.e. meteorological, hydrogeological, socio-economic and a countermeasure component) (Connor and Hiroki, 2005; Balica et al., 2009). A similar index is the country level physical and community risk index for earthquakes and floods in the Asia-Pacific region by Daniell et al. (2010). Kannami (2008) developed a country-based flood risk index (FRIc), based on the Pressure and Release (PAR) model.
- UNDP's 2004 Disaster Risk Index (DRI) is an index that aims to explain the role of vulnerability for different risk levels or different numbers of post-disaster fatalities between countries with a given level of physical exposure to three types of natural disasters (i.e. earthquakes, tropical cyclones, floods and, in more recent versions, droughts) (UNDP, 2004; Birkmann, 2007; Peduzzi et al., 2009). Indicator selection focuses on allowing comparison between countries and hazard types (DRI indicators are hazard specific) (UNDP, 2004; Birkmann, 2007).
- Yücemen et al. (2004) developed a multivariate-statistics analysis to assess "the seismic vulnerability of low- to mid-rise reinforced concrete buildings". The six selected indicators are all engineering-based using expert judgment and observations. The model uses five discrete damage states ranging from *none* to *collapse*. It calibrates this based on empirical damage seen in historical events in Turkey.

### 2.2.2    Vulnerability curve models

- The vast majority of flood- and earthquake vulnerability assessment models are based on damage functions or fragility curves that relate the (mostly-) physical indicators described in Sect. 2.1 with hazard parameters (Douglas, 2007). In flood damage models, vulnerability is commonly calculated by relating flood depth to building or land-use type using vulnerability curves per exposed building- or land-use type. These curves provide estimates of potential damage. Occasionally, other hazard parameters such as velocity and duration are added (Merz et al., 2010; Jongman et al., 2012). Unlike most other hazard type risk assessments, earthquake risk assessments traditionally use fragility curves as a vulnerability, or expected damage, measure, in which probabilistic damage to, for example, buildings is related to a hazard parameter such as ground shaking intensity (Douglas, 2007). In this study, we grouped fragility curve-based models with other curve-based models. Several examples are:

- The HAZUS-Multi Hazard (HAZUS-MH) is a risk model developed by the National Institute of Building Sciences for the Federal Emergency Management Agency (FEMA) in 1997. It addresses four types of natural hazards (i.e. coastal storm surge, earthquakes, river flooding and windstorm damage) and estimates both direct and indirect economic losses (Kircher et al., 2006; Remo and Pinter, 2012). HAZUS-MH' earthquake component uses analytically derived damage curves (Spence et al., 2008). These curves are designed for US buildings, which complicates application to different parts of the world. The flood hazard component addresses riverine and coastal flooding (Scawthorn et al., 2006a; Nastev and Todorov, 2013) and uses more than 900 damage curves mostly derived from FEMA (Scawthorn et al., 2006a). The flood vulnerability component addresses susceptibility to damage, loss and injuries. The HAZUS model, encompassing the capacity spectrum method, has been applied to various locations globally in various software packages including an Australian calibrated methodology – EQRM (Robinson et al., 2005), SELENA (Norway, India and other locations) (Molina et al., 2010), HAZTaiwan (Loh etal., 2000).

- There are many flood risk models that use vulnerability curves, such as Hazus-MH, the Multi-Coloured Manual (MCM), GLOFRIS, the Damagescanner and the European Flood Awareness System (EFAS) (Meyer and Messner, 2005; Jongman et al., 2012; Ward et al., 2013). The MCM by Penning-Rowsell et al. (2010) is the most advanced curve-based flood damage assessment method in Europe (Jongman et al., 2012). Similar to HAZUS-MH, the MCM is an object-based model where buildings are classified based on building usage (i.e. residential, commercial and industrial) (Meyer and Messner, 2005), however it uses absolute depth-damage curves to relate damage in British Pounds to water depth. The MCM does not include indirect flood damages but it does account for short and long flood durations (Meyer and Messner, 2005; Jongman et al., 2012).

- The Prompt Assessment of Global Earthquakes for Response (PAGER) was developed by the U.S. Geology Survey (USGS) (Wald et al., 2008; Jaiswal et al., 2011). PAGER incorporates three different approaches for assessing vulnerability, i.e. empirical, semi-empirical and analytical. In predicting future vulnerability, the empirical approach uses historic country-level earthquake data and calibrates casualty rates to develop regression parameters (Jaiswal et al., 2011). In the semi-empirical and analytical models, the building inventories together with data on each structure's occupancy type, intensity-based vulnerability (building collapse rate) and the fatality rate are used to derive fatality functions (Jaiswal et al., 2011). In the analytical approach, the same building inventories and the occupancy types as in the semi-empirical approach are used. However, vulnerability (collapse rates) are based on engineering considerations (such as the HAZUS capacity spectrum based approach) (Wald et al., 2008). index-based

## 3    Results and discussion

In this section, we show and discuss the results of the literature review. In Sect. 3.1, a comparison between physical and social vulnerability indicators is presented. Next, in Sect. 3.2, we compare earthquake and flood vulnerability models.

### 3.1 Physical versus Social Vulnerability Indicators

Tables 1 and 2 respectively present an overview of the different physical and social vulnerability indicators, whereby the columns distinguish between curve- and index-based vulnerability assessments. In the rows, each indicator is further sub-divided into the indicator classes provided in Sect. 2.1. Indicators have been briefly described with their unit and scale of the exposed elements they refer to (Ob: object; Agg: aggregated; Com: combination of both). We also show the geographical scale of the application of the indicators and their models (L: Local; R: Regional; N: National; G: Global). The numbers behind each indicator provide examples of papers using that particular indicator.

### 3.1.1 Physical Indicators

Several physical indicators are used in both domains, such as building material, number of stories, accessibility of roads, etc. However, there are also indicators used in one domain but less-frequently or not at all used in the other. Table 1 supports the claims made in Sect. 2 that earthquake vulnerability assessments make use of highly detailed indicators at a building level, as they distinguish the number of stories, occupancy class, and building material. For example, Daniell (2015) provides a global review of country-level seismic-building codes from 1900 to 2013). Flood vulnerability assessments have seen a recent transition from focusing on traditional flood protection measures which aim to decrease the flood probability for an area to building-specific resilience measures (Ashley et al., 2007; Naumann et al., 2011). One example where this has been done is a study by Nikolowski (2014) which provides an overview of different ranges of building age and their flood vulnerability; structural (load carrying) and non-structural (mechanical) components; roof types; and building maintenance factors. For flood, vulnerability of building- or land-use types are often related to flood hazard indicators such as flood depth or flood velocity to estimate potential losses (e.g. Roos 2003; Barroca et al., 2006).

For earthquakes, fragility curves are used to relate building damage to the amplitude of ground shaking (Birkmann and Wisner, 2006; Calvi et al., 2006; Douglas 2007). Detailed vulnerability indicators for buildings are described by Daniell et al. (2012a) and for infrastructure in Daniell (2014). Davidson and Shah (1997) argue that some of these indicators, such as maintenance, previous damage, and retrofitting affect the physical earthquake vulnerability but that data to measure this is hard to obtain. These indicators can be measured using time-consuming processes such as using cadastre or census data, or by sampling the buildings of a neighbourhood or city. For example, Steimen et al. (2004) assessed 10% of the building stock in the city of Basel (Switzerland). Rashed and Weeks (2003) include lifeline and infrastructure as well as building related indicators (i.e. transportation and utility lifelines, square footage, inventories, cost of building repair). Menoni and Pergalani (1996) include a building usage classification and account for the nearby existence of hazardous plants.

**Infrastructure and lifelines indicators**

Infrastructure and lifeline indicators are used both in earthquake and flood vulnerability assessments, for example inHAZUS-MH. Atzl and Keller (2013) provide a framework which links social vulnerability to critical

infrastructure and create indicators at the individual level for infrastructure-specific social vulnerability of commuters in Stuttgart (e.g. travel distance, availability of alternative transport, and number of available public transport lines). As shown in Table 1 and as argued in other work (Miletti, 1999), there are   fewer flood vulnerability assessment studies including infrastructure related indicators compared to earthquake vulnerability assessments. Keller and Atzl (2014) add to the existing body of experimental research by assessing the causal relation between extreme precipitation events and the impacts on German infrastructure using an explanatory approach. In other studies, earthquake vulnerability assessment models are occasionally adopted in flood vulnerability models to address infrastructure risk (Merz et al., 2010). However, the knowledge gap continues to exist and there is a need for further research (Keller and Atzl (2014).

Traditionally, there is a strong focus within earthquake vulnerability studies on indicators of *utility and essential facilities lifelines* (i.e. utility systems such as electricity, telecommunication, potable and waste water, and infrastructure) (Menoni et al., 2002; Menoni et al., 2007). Frequently used lifeline vulnerability indicators measure the length and accessibility of lifelines, such as a road (e.g. Penning-Rowsell et al., 2010; Peng, 2012). An extensive and highly detailed overview of the lifeline indicators that Menoni et al. (2002) used in earthquake vulnerability assessments, and in fragility curves in particular, is provided by Pitilakis et al. (2014). Flood vulnerability assessments use similar lifeline indicators such as the physical aspects of road networks (e.g. Barroca et al., 2008).

**Building structural and occupancy indicators**
The need for detailed earthquake loss estimations for the insurance and re-insurance industry has advanced the development of detailed, object-based (e.g. building-level), vulnerability assessment models (Spence et al., 2008). An extensive overview of earthquake loss estimation models (ELE) and their respective definition of vulnerability classes has been provided by Daniell (2012a and 2014). As part of earthquake vulnerability assessments' emphasis on individual building characteristics, the building age is an important indicator. Generally, the influence of building age on earthquake vulnerability levels is twofold: (a) with aging comes deterioration of building materials; and (b) more recently constructed buildings have more often been subjected to improved building codes (Cochrane and Schaad, 1992; Bommer et al., 2002)..

Another example of earthquake vulnerability's focus on buildings and the inclusion of more detailed building related indicators, is the building material type indicator (e.g. wood, steel, concrete, masonry or mobile homes). Building material type is a crucial factor in determining a building's ability to resist ground shaking and is used in many models such as HAZUS-MH (Kircher et al., 1997 and 2006; Bommer et al., 2002; Nastev and Todorov, 2013). It should be noted that a specific building type can have opposing impacts on earthquake versus flood vulnerability. For example, wooden houses tend to be more vulnerable to flooding than stone houses, but for earthquakes generally the opposite holds (Doğangün et al., 2006; Messner and Meyer, 2006).

Another important factor is the number of stories. For flooding, multi-story buildings are generally susceptible to a lower damage fraction than single-story buildings (Merz et al., 2010). Moreover, people can evacuate to higher floors in case of a flood, reducing the number of fatalities. For earthquakes, however, multiple floor buildings can have a higher vulnerability depending on the frequency content of the ground motion (which influences the dynamic response of structural systems) of the earthquake. Moreover, for earthquakes there are more complicating factors, for example *enforced seismic design codes* and the *type of energy-wave as a result of an earthquake* influence the correlation between building height and vulnerability (Rossetto and Elnashai, 2003).

Papathoma-Köhle et al. (2011) discuss the difficulties in assessing vulnerability of the built environment for different Alpine hazards, including floods. They conclude that most vulnerability assessment methods are quantitative. For floods, damage curves linking water depth to building damage are well-developed for Europe and similarly developed countries. However, these curves do not apply to other parts of the world due to differences in building material and construction type (Papathoma-Köhle et al., 2011).

Within flood vulnerability assessments, some research have been conducted regarding non-structural damages and disaster risk reduction measures (e.g. building regulations pushing for flood-proofing) to reduce building content damages (Dawson et al., 2011). However, rather than using a separate indicator, several models include content damage by adjusting the shape of the damage curve or changing maximum damage values. HAZUS-MH uses a 0.5 factor for estimating residential content damages in relation to structural damages (Scawthorne et al., 2006) and this factor has also been used by other studies (e.g. Penning-Rowsell et al., 2010; de Moel et al., 2014). The Damagescanner, a curve-based flood vulnerability assessment model, accounts for three types of flood-proofing measures (i.e. wet-proofing, dry proofing and a combination of the two) in assessing future potential for damages by adding damage reduction factors (0-1) (Poussin et al., 2012).

### Environmental indicators

As shown in Table 1, environmental indicators consist of two aspects: the proximity to contaminating sites (e.g. Menoni et al., 2002; Damm 2009) and the susceptibility and vulnerability of the environment captured in indicators such as *types of vegetation*, *soil erosion potential* and *soil quality* (e.g. Barroca et al., 2008; Balica et al., 2009; Damm 2009). The latter appear to be more often taken into account as part of flood vulnerability assessments.

### 3.1.2    Social Indicators

Tate (2012) argues that the social vulnerability index is the social equivalent of the quantitative physical vulnerability assessment. In these indices, demographic data is often used to describe social, economic, political and institutional vulnerability. However, since there is a lack of systematic evaluation of how social vulnerability indices are constructed, little is known about how well these social vulnerability indices perform (Tate 2012). Tate (2012) concludes that most studies only provide limited justification for the inclusion of specific indicators. He argues that researchers should give more thought as to which social indicators to include as well as their statistical

properties.

To assess exposure differences to flooding and whether those who are most exposed also have the highest social vulnerability, de Sherbinin and Bardy (2015) apply their social vulnerability index using different sets of indicators to New York and Mumbai. Their method build on earlier work by Cutter et al. (2003) and the IPCC

Special Report on Extreme Events Framework (IPCC 2012). Inclusion of indicators differed for the two cities and was often dependent on data availability and applicability to the case study (de Sherbinin and Bardy, 2015).

There are fewer differences between the types of social vulnerability indicators used in flood and earthquake vulnerability assessments, compared to the differences found for physical vulnerability indicators. However, from

the literature review it appears that social indicators are more often used in flood vulnerability studies than earthquake vulnerability studies. Examples of earthquake social vulnerability indicators are: population density (Menoni and Pergalani, 1996; Peng, 2012); household education level (Duzgun et al., 2011; Schmidtlein et al., 2011); shelter demand (e.g. measured using 'perception of population to leave their homes' indicator); health impact related vulnerability as part of SYNER-G's socio-economic vulnerability component (Pitilakis et al.,

2014); and household and population structure as used in GEM's socio-economic vulnerability index (Khazai et al., 2014a). For flooding, similar indicators are used, such as population density (Balica et al., 2012), education level (Cutter et al., 2006) and GDP (Balica et al., 2009; Ferreira et al., 2011) and long-term sickness (Tapsall, 2002).

**Demography**
Flood and earthquake studies both use very similar demographic indicators, such as the identification of weaker groups in society based on age (e.g. those younger than 5 and older than 65 years) and other indicators such as wealth, ethnicity, family structure, and disabled people (Cutter et al., 2003; Schmidtlein et al., 2008; Fekete, 2009; Blaikie et al., 2014).  The high importance of *age* as an indicator of social vulnerability is also supported by Rufat

et al. (2015). A household's socio-demographic status plays a crucial role in their social vulnerability and their ability to prepare for future disasters. It is often measured using indicators such as education level and percentage of population living in poverty (Cutter et al., 2003; Koks et al., 2015b; Rufat et al. 2015). In a study of the Rijnmond region in the Netherlands, Koks et al. (2015b) simulate the spatial distribution of social vulnerability, using indicators such as ethnicity, age group (elderly) and fiscal income.


Within earthquake research, population-related indicators are used to establish the number of (vulnerable) people present in offices, residences or schools, which is often influenced by the time of the day. This particular focus of the influence on timing and building occupancy is common in earthquake vulnerability assessments (e.g. Lomnitz, 1970; Coburn and Spence., 1992; Ara, 2013). Whilst prominent in earthquake research, these aspects are

not taken into account in flood vulnerability assessments. As shown in Table 2, within flood vulnerability assessments there are fewer social indicators used than for earthquake vulnerability assessments but with many

studies using similar indicators, social indicator usage appears to be more perfected (e.g. Rufat et al., 2015). For earthquake vulnerability assessments, this appears to be less the case, and more different types of indicators are used.


### Awareness

Furthermore, some flood vulnerability assessments use preparedness indicators, such as flood risk awareness, past experiences, and the effect of media exposure on peoples' risk perception (Rufat et al., 2015). Research has shown that previous experience with a disaster (e.g. property damage or loss and personal distress) has a strong

correlation with how people prepare for a next disaster (Lindell and Perry, 1992). In a study of flood preparedness in Dresden, Kreibich and Thieken (2009) show that there is a strong correlation between flood risk awareness and improvements in flood levels of individual households. On the other hand, more recent studies with regards to earthquake awareness found a lower correlation between past earthquake experience and awareness, but noted a relationship between education and awareness and preparedness (Rüstemli and Karanci, 1999; Shaw et al., 2004).

Also with regards to the impact of social development and welfare levels on vulnerability, flood assessments use comprehensive indicators more often than earthquake vulnerability assessments, such as: education level and literacy rate; technological development (e.g. ownership of tv, radio, phone, etc.); and other means of connectivity (e.g. Akukwe and Ogbodo, 2015). Another difference is the usage of a warning-time indicator. Although there is still debate about the inclusion of such an indicator (e.g. Merz et al., 2010), flood vulnerability assessments

occasionally include a warning-time indicator (e.g. Penning-Rowsell et al., 2010; Scawthorn et al., 2006b). For flooding, it has been shown that when the warning time is increased by more than two hours, damage can be reduced by more than 10% (Penning-Rowsell et al., 2010; Messner and Meyer, 2006). However, warning-time is not an indicator used with regard to earthquakes, where, due to the nature of earthquakes, the warning time can be a matter of only a few seconds (Nakamura and Saita, 2007).


### Socio-Economic Indicators

Within the sub-category of socio-economic indicators, flood and earthquake vulnerability assessments both use similar income-related indicators such as GDP. Earthquake vulnerability assessments also tend to take sector dependency of a community into account, generally measured through the percentage of people employed in one

sector. It has been shown that single-sector dependency increases a community's vulnerability (Cutter, 2003). From Table 2 it also appears that flood vulnerability assessments tend to take more indicators of welfare and social security levels into account than earthquake vulnerability assessments. Khazai et al. (2014) argue that for earthquakes, most often social vulnerability is integrated as a linear consequence function of physical damage (e.g. building damage causing casualties). For earthquake vulnerability, the index-based SYNER-G framework

designed by Khazai et al. (2014) integrates physical and social indicators where both are assumed to be a direct function of hazard intensity, physical vulnerability and social vulnerability of the at risk population. For example, the expected number of post-disaster homeless people depends not only on the number of damaged buildings but also socio-economic indicators. Khazai et al. (2014) focus on including socio-economic indicators that can be

quantified and harmonized at an EU-level and urban scale which led to the inclusion of more often used indicators
such as household tenure (proportion of households living in self-owned or rented housing). Socio-economic
indicators use aggregated data and are mostly used in index-based vulnerability assessments rather than in curve-
based vulnerability assessments.

**Institutional and Political Indicators**

Table 2 shows that it is more common for flood vulnerability assessments to include indicators related to zoning
and land-use planning. For floods, indicators related to increasing resilience such as urban planning institutions
(Balica et al., 2009) and investments in precautionary measures (Connor and Hiroki, 2005) are occasionally
considered in assessing social vulnerability. For earthquakes, it appears that fewer models take governance-related
indicators into account; such as *political stability* (GEM, 2016) and crime rates (Burton and Silva, 2014). Non-
hazard specific vulnerability assessment models such as the Disaster Risk Index (DRI) by Peduzzi et al. (2009)
use Transparency International's corruption perception index (CPI) as an indicator of corruption, and the
Prevalent Vulnerability Index (PVI) incorporates a governance index which the following six indicators: Voice
and Accountability; Political Stability; Absence of Violence; Government Effectiveness; Regulatory Quality; Rule
of Law; and Control of Corruption (Cardona and Carreño, 2011; IPCC, 2012).

**3.2    Vulnerability models**

**3.2.1    Curves versus index-based vulnerability assessments**

Our study supports the claims that for both earthquake and flood vulnerability models, a large suite of well-
developed vulnerability damage curves exists (Douglas, 2007). For assessing social vulnerability, aggregated data
as well as index-based vulnerability assessments are more commonly used for both floods and earthquakes than is
the case for physical vulnerability. For both floods and earthquakes, these index-based vulnerability assessments
tend to incorporate demographic indicators much more frequently than assessments based on vulnerability curves.
Examples of indicators used in index-based vulnerability assessments are to find relationships between:
*inventories of building square footage* and *inventories of building value* and *reported earthquake losses as a*
*percentage of modeled exposed GDP* (Rashed and Weeks, 2003). Or in flood modeling: *reported fatalities as a*
*percentage of modeled exposed population* (Jongman et al., 2015). Rufat et al. (2015) argue that in recent years
indices have become the main tool used to assess social vulnerability to flooding.

Developing meaningful vulnerability indices is difficult, and complex interrelations between vulnerability and
hazard or damage are often represented in simple indices (Cutter et al., 2003; Birkmann, 2007; Chang et al.,
2015). On a positive side, empirical data on losses, required to relate vulnerability indices to those losses, have
been improved over the last 10 years. However, more data are needed and loss data on (extreme-) hazard events
are scarce. New global databases of empirical natural disaster loss data include CATDAT (Daniell, 2009), the
International Disaster Database (EM-DAT), and UNISDR's Disaster Information Management System

(DESinventar). These databases provide useful quantitative input to risk- and vulnerability assessment studies such as PAGER and GEM (Jaiswal et al., 2011; Dell'Acqua et al., 2013; Silva et al., 2014a and 2014b).

For physical vulnerability, earthquake vulnerability assessments show a much more important concentration on buildings and object-level vulnerability curves. Table 1 shows that, for earthquake vulnerability assessments, indicators for utility lifeline vulnerability are commonly employed as part of index-based vulnerability assessments. Only few studies on flood vulnerability have similarly addressed utility lifeline indicators. For example, Barroca et al. (2006) incorporate flood lifeline indicators such as: physical aspects of utility lifelines including *energy networks*, *physical aspects of urban lighting*, *heating*, and *water supply networks*.

### 3.2.2    Spatial versus temporal aspects

Hinkel (2011) explains that, using indicators, changes in vulnerability can be assessed either over time for a set entity (e.g. an administrative level or a group of people) over time or in space at a set time (i.e. between geographic entities). Therefore, we also compared the different vulnerability models for scale and temporal aspects. The importance of incorporating both temporal and spatial scales in vulnerability models has been addressed by many studies (e.g. Cutter et al., 2003; Barroca et al., 2006; Zevenbergen et al., 2008; Fekete et al., 2010; Jongman et al., 2015).

**Spatial scale**

An important aspect of vulnerability assessments is their spatial scale (Cutter et al., 1996). Vulnerability assessment models can be applied on different spatial scales (high versus low resolution) and using different data types (object versus aggregate, or raster, based). This is often dependent on data availability: particularly for social vulnerability indicators it is challenging to find high quality social vulnerability data for measuring those indicators at a local level (e.g. de Sherbinin and Brady, 2015). For different hazards, Birkmann (2007) reviewed indicator usage across three global risk models and a local approach. It appears from this study that downscaling vulnerability indices is very difficult due to data scarcity. Therefore, flood vulnerability assessments generally have a high level of spatial aggregation, often using land-use data to represent exposure. This is also recognised in the literature (e.g. Comfort et al., 1999; Barroca et al., 2006, Zevenbergen et al., 2008), where it has been acknowledged that future flood vulnerability studies should receive more attention and be more available to stakeholders at a local or city level. Some flood assessment tools, however, such as the Flood Vulnerability Analysis Tool (FVAT) (Barroca et al., 2006 and 2008), provide indicators that are available at a local level. Balica et al. (2009) developed their Flood Vulnerability Index (FVI), which is applicable at different spatial scales such as river basin, sub-catchment and urban areas.

Indicators used in vulnerability curve methods for earthquakes seem to have more detail (e.g. *building maintenance level*, *roof type* and *height*) as compared to the flood models. For both vulnerability curve-based as well as index-based vulnerability assessments, earthquake vulnerability assessments have a very strong focus on

individual buildings, their construction and structural characteristics, as well as their ability to resist seismic
tension as a primary cause of damage and casualties. HAZUS-MH and the Multi-Coloured Manual by Penning-
Rowsell et al. (2010) are among the few flood vulnerability models that are curve-based and developed at an
object level (Jongman et al., 2012). On the other hand, the general approach in earthquake modeling is to
categorize the general building stock into small groups whose characteristics (e.g. strength, weight, construction
material, height, construction quality, and age) create similar seismic responses (Ventura et al., 2005). Building
classification systems are used to group buildings based on these characteristics. Next, damage functions are
created based on the estimated damage due to ground motion for each building class (Ventura et al., 2005).

Some of the indicators used in earthquake studies are also used in flood studies (e.g. number of stories, building
height and age). However, flood risk assessment models are often designed at an aggregated land-use class level
whereas earthquake risk assessments often make use of fragility curves which mainly focus on objects (often
buildings).

The indirect economic impacts of a local flood on the regional and national economy can be substantial, which
underscores the necessity of understanding indirect flood vulnerability (Zevenbergen et al., 2008; Balica et al.,
2009). This indirect factor is currently either ignored or modeled in a rather simplistic way. For example, in
HAZUS-MH it is modeled as a fraction of the direct losses. However, new flood research using economic
methods shows indirect losses can be substantial and widespread (Koks et al., 2015a).

In terms of upscaling social vulnerability indicators, Fekete et al. (2010) recognize the importance and lack of
flood vulnerability studies that account for cross-scale interactions. Some demographic indicators collected at a
household or individual level can be scaled up. However, social indicators such as *power structures* cannot,
because they are not "significantly linked to the structure of a household or person" (Fekete et al., 2010). Koks et
al. (2015b) focus on social vulnerability and found that in future flood risk scenarios there is a clear spatial
clustering of socially vulnerable groups measured through social vulnerability indicators such as: age, fiscal
income, and ethnicity. Other studies have used spatial analysis techniques to identify clusters of vulnerability
(Rashed and Weeks, 2003; Rashed et al., 2007).

In flood assessment studies, it is more common to use aggregated exposure data, such as land-use data from
satellite observations as a basis for estimating vulnerability at the river basin, country of continental scales
(Jongman et al., 2012; de Moel et al., 2015). Land-use data often replaces building scale data, because: (a)
building data is not available at larger scales; and (b) computational efforts are too challenging using detailed
exposure data at these scales. Examples of such flood damage models that are land-use based, are: the
DamageScanner (e.g. Klijn et al., 2007), FLEMO (e.g. Apel et al., 2009), and the JRC Model (Huizinga, 2007).
We refer to Jongman et al. (2012) for a comparison among different flood damage model assessments.

**Temporal Scale**

An interesting aspect of earthquake and flood vulnerability assessments is the extent to which they consider temporal scales in vulnerability, for example through the implementation of building codes or other mitigation policies, land-use change, demographic changes such as population growth, and social- and economic changes (e.g. Zevenbergen et al., 2008). Understanding flood vulnerability over time is crucial in examining past, current and future fatalities and losses (Jongman et al., 2015), and can significantly improve a risk managers' ability to more efficiently implement mitigation measures (Birkmann, 2007; Schmidtlein et al., 2011). Therefore, the focus has shifted to assessing vulnerability over time (Jongman et al., 2015), but knowledge gaps continue to exist (Connor and Hiroki, 2005; McEntire, 2005; Birkmann, 2007; Cutter et al., 2008; Balica et al., 2012; Mechler and Bouwer, 2014; Jongman et al., 2015; Koks et al., 2015b).

Chang et al. (2012) studied temporal changes in the seismic risk of Vancouver (Canada). Using a M7.3 earthquake scenario, this study concludes that despite increasing exposure (the population of Vancouver doubled over the course of the 35-year study period from 1971 to 2006), the estimated 2006 casualties remained equal to the estimated number of casualties in 1971. They conclude that the decrease in the per capita casualty ratio is mainly due to improvements in building codes and construction changes. Daniell (2015) provides a global overview of seismic-building codes implemented from 1900 until 2013, which shows that the number of countries with a seismic code or zonation has increased (although it should be noted that currently less than 50% of the building stock is covered by a building code). There are several challenges in incorporating temporal scales in earthquake vulnerability assessments. Earthquake vulnerability research mainly focuses on predicting the ability of the (current) building stock to withstand ground shaking. It has been shown that the selection of a building inventory very strongly influences earthquake vulnerability. Faccioli et al. (1999) explain that there are some significant difficulties involved in creating a reliable building inventory for earthquake scenario studies. Steimen et al. (2004) therefore underscore the necessity of uncertainty analysis in earthquake scenarios and building vulnerability estimates. A country-level method for the development of an earthquake risk exposure model for buildings is introduced by Gunasekera et al. (2015).

Another problem in using earthquake scenarios to address temporal changes in vulnerability is the lack of confidence in estimating the location and strength of an earthquake (Faccioli et al., 1999). Menoni et al. (2002) developed a tool to study earthquake event scenarios for lifelines to estimate both the physical and organizational failures originating from lifeline systems. Summarizing, it appears that temporal changes regarding earthquake risk mainly focus on temporal changes in exposure rather than vulnerability. Duzgun et al. (2011) developed an earthquake vulnerability assessment framework for urban areas, which "enables decision-makers to monitor temporal and spatial changes in the urban environment due to implementation of risk reduction strategies".

There are several papers that include the impacts of temporally changing factors that are not specific for a particular hazard type on flood vulnerability, such as population growth (e.g. Hall et al., 2005; Ferreira et al.,

2011; Rojas et al., 2013). Hall et al. (2005) look at changing flood risk in England and Wales using a scenario-based approach for 45 and 75 years into the future with changing climate and socio-economic conditions and conclude that economic vulnerability (e.g. increasing infrastructure vulnerability) combined with climate change effects will increase by 2080 causing an increase in flood risk. Hall et al. (2005) use the social flood vulnerability indices as introduced by Tapsell et al. (2002), which constitute an aggregated measure of population vulnerability. Rojas et al. (2013) also acknowledge the lack of studies that have considered the quantification of adaptation measures. In a comparative study, Rojas et al. (2013) look at a *no-adaptation* versus an *adaptation* scenario of future flood risk mitigation (accounting for socio-economic developments and changing population density). Ferreira et al. (2011) focus too on social- and economic indicators (e.g. GDP, GINI coefficient, domestic credit to the private sector, expressed as a percentage of GDP, indicators for corruption, bureaucratic quality, law and order, democratic accountability, government stability, ethnic tensions, and religious tensions) in their study of flood adaptation.

Although vulnerability is usually assumed to be constant, often due to difficulties in accounting for changing vulnerability, several studies have shown the impact of vulnerability reducing measures on risk reduction (Mechler and Bouwer, 2014; Jongman et al., 2015). In a case study of the Meuse, Poussin et al. (2012) use the Damagescanner to show that annual flood risk may increase with 185% over the period 2000 to 2030 due to both land-use and climate changes. However, the study shows that implementing adaptation strategies such as spatial zoning and other vulnerability mitigating measures, including dry- and wet-proofing of buildings, do decrease future risk levels with the relative risk reduction ranging from 10% to 40% depending on the specific measure (Kreibich et al., 2015; Kreibich and Thieken, 2009; Poussin et al., 2012). In a study of the impacts of land-use and climate changes on flood risk of unembanked areas of Rotterdam, De Moel et al. (2014) also find that building-level mitigation measures (e.g. elevating buildings) reduce future flood risk.

## 4    Conclusions and Recommendations

This cross-discipline study allowed us to obtain lessons from earthquake and flood vulnerability assessments that could be used for advancing risk assessments in both fields. In general, indicators used in earthquake and flood vulnerability assessments have substantial differences. Below we summarize our main findings, which are also shown in Table 3. The numbers refer to the conclusions in that table.

1. While flood vulnerability assessments exist at different spatial scales, flood vulnerability research could benefit from improving assessments at the more local and object scale.

2. This difference between object- versus aggregate scale vulnerability assessments strongly relates to the focus of earthquake vulnerability assessments on physical vulnerability. Despite the differences in application, the physical (i.e. building) aspects of flood vulnerability assessments could be improved by incorporating earthquake vulnerability assessment methods and indicators, specifically for an object (building) based approach. For example, the development of building material based approaches for flood vulnerability assessments lacks behind that of earthquakes. Combined with an object-based

approach this could push forward the development of depth-damage curves that make use of building material at an object-level.

3. Another poignant difference appears to be that flood vulnerability assessments more often take into account indicators related to risk awareness and precautionary measures at a governmental as well as individual level, compared to earthquake vulnerability assessments. This is something where earthquake
vulnerability assessments could learn from flood vulnerability assessments.

4. Flood vulnerability assessments tend to use more precise indicators of social vulnerability than earthquake vulnerability assessments, and flood vulnerability assessments more often include indicators related to welfare and social security levels.

5. However, earthquake studies do tend to incorporate aspects of local economic-sector dependent
vulnerability more often than is the case for floods.

6. Another difference is the use of a timing indicator used in earthquake modeling, which shows where people are located throughout the day. Timing and an estimate of where people are during the day could be a useful factor for improving flood risk assessments. At the same time, earthquake modelling could benefit from modelling evacuation patterns as done in flood assessments.

7. Flood assessment models examine the impacts of changing exposure over time on vulnerability more often than earthquake assessments, for example due to the implementation of adaptation measures. One way of improving this aspect of earthquake vulnerability assessments, would be to better incorporate indirect economic loss assessments from natural disasters such as recently published for flood risk. This would benefit and enable more analytical (rather than judgment based) future mitigation and adaptation
studies.

One of the issues encountered was that not all studies mention specifically which indicators they use for their vulnerability assessment. Some studies mention the categories or theoretical indicators they look at but do not list the 'measurable indicators' used explicitly. Furthermore, studies that take into account vulnerability in their risk
assessment but do not explicitly model the vulnerability component itself, have been excluded from this study, and we have only assessed a selection of the wealth of models that is available. Another complicating factor comes from the difference in spatial scales used. Flood vulnerability indicators are used in case studies with a less detailed spatial scale compared to earthquake vulnerability indicators, which are generally applied to smaller scale case studies. In trying to obtain cross-discipline lessons this forced us to include multiple scales and compare
across multiple scales (from local to national). Furthermore, due to the challenges in assessing vulnerability (as explained in section 1), most risk assessment models focus on physical vulnerability, as its indicators are more easily quantifiable, which might lead to them being over-represented in our study. Finally, we acknowledge the limited scope of our study, however the focus on risk assessment models using quantifiable indicators allowed us to better understand how flood risk assessment models can be improved.


In general, we advocate cross-disciplinary learning between the earthquake and flood risk modelling communities.

An ideal flood vulnerability method encompasses a balanced mix of the two different components: physical and socio-economic related indicators and attempts to move towards an object scale approach. Furthermore, it is very important to increase understanding of the interaction between flood and earthquake vulnerability and how these can be assessed simultaneously in a risk assessment. Some factors can have positive effects on reducing vulnerability of, for example, floods while simultaneously having negative impacts on earthquake vulnerability. For example, building houses on stilts can be very beneficial in decreasing flood vulnerability while increasing earthquake vulnerability. This calls for more collaboration between the two research communities. More studies are looking into cascading events. We recognize this as an emerging field, and believe this field will benefit from further comparative research, involving more models and methods.

**Competing interests**

This research was funded by the Netherlands Organisation for Scientific Research (NWO) via VICI grant 453.14.006. PJW received funding from NWO in the form of a VIDI grant (016.161.324). The authors declare that they have no conflict of interest.

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

| Vulnerability indicator category | FLOOD VULNERABILITY | | EARTHQUAKE VULNERABILITY | |
| --- | --- | --- | --- | --- |
| | Vulnerability curves | Index | Vulnerability curves | Index |
| *Infrastructure and lifelines* | • Material and segment length (Ob, L) [12,27] <br> • Traffic volumes, extra travel times (Ob, L) [23] <br> • Material, anchored (Y/N) and segment length (Ob, L) [12, 27, 28] <br> • Structure, occupancy, quality (Ob, L) [12, 27,28] | • Location, availability and length of roads (Com, L) [4] <br> • Location and length of utility lifelines (Com, L) [4] | • Material and segment length (Ob, L) [11, 16] <br> • Material of supporting system of tunnels (Ob, L) [24] <br> • Shape and depth of tunnels (Ob, L) [24] <br> • Bridge design type (e.g. single versus multiple span) (Ob, L) [24] <br> • Material, anchored (Y/N) and segment length (Ob, L) [11, 16] <br> • Natural gas pipeline material and construction types (Ob, L) [24] <br> • Structure, design level, occupancy class, construction quality factor (Ob, L) [11, 16] | • Location and availability of transportation facilities (Com, R) [22, 26] <br> • Accessibility of utility lifeline (Ob, L) [19] <br> • Maintenance of utility lifeline (Ob, L) [19] <br> • Age of utility lifeline (Ob, L) [19] <br> • Closeness one utility to another (Ob, L) [19] <br> • # Lifelines on bridges and viaducts (Ob, L) [19] <br> • Accessibility of essential facilities (Ob, L) [19] |
| *Buildings* | • Building structural types (Ob, L) [12, 22, 23, 26, 27, 31] <br> • # of stories (Ob, L) [12, 23, 27, 28, 31] <br> • Building height (Ob, L) [12, 23, 27, 28, 31] <br> • Building age (Ob, L) [23] <br> • Foundation type (Ob, L) [12, 27, 28] <br> • Building occupancy (Ob, L) [12, 22, 23, 26, 27] | • Quality of building structure (Agg, L) [1, 20] <br> • # of stories (Agg, L) [20] <br> • Floor space of building (Agg, L) [20] Building occupancy class (Ob, L) [3, 4, 20] | • Building structural types (i.e. material) (Ob, L-G) [6, 8, 11, 13, 15, 16, 18, 21, 25] <br> • # of stories (Ob, L-G) [11, 13, 16, 18, 21] <br> • Building height (Ob, L-G) [6, 11, 13, 16, 18, 21] <br> • Building age (Ob, L) [10, 14] <br> • Roof type (Agg, G) [13] (Com, L-G) [10, 18] <br> • Building maintenance (Ob, L-R) [13, 17] <br> • Building configuration (Ob, L) [16, 21] <br> • Wall structural type (Com, L-G) [10, 18] <br> • Date of construction retrofit (Com, N) [6] <br> • Lateral load-resisting system (Com, N) [6] <br> • Building occupancy (Ob, L-G) [5, 6, 11, 16, 18, 29] | • # Stories (Agg, L-R) [30] <br> • # Stories above ground level (Agg, L-R) [30] <br> • Building height (Agg, L-R) [30] <br> • Roof type (Agg, L-R) [30] <br> • % of buildings in need of large repairs (Agg, N) [7] <br> • Soft story index (ratio of the ground story height to the first story height) (Agg, L-R) [30] <br> • Normalized redundancy score (Agg, L-R) [30] <br> • Min. norm. lateral stiffness index (Agg, L-R) [30] <br> • Overhang ratio (the floor area beyond outer frame / area ground fl (Agg, L-R) [30] <br> • Completed buildings in new constructions per 800 population (Agg, N) [7] |

| Environmental | | • Proximity to contaminating sites (Agg, R) [9]<br>• Types of vegetation (Agg, R) [2]<br>• Soil erosion potential (Agg, R) [9]<br>• Soil quality (Agg, R) [2, 4] | | • Proximity to contaminating sites (Ob, L) [19] |
|---|---|---|---|---|

**Table 1: Overview of physical earthquake and flood vulnerability assessment indicators.**

**Selected references:**

**[1] Akukwe and Ogbodo, 2015**

**[2] Balica et al., 2009**

**[3] Barroca et al., 2006 (FVAT)**

**[4] Barroca et al., 2008**

**[5] Bommer et al., 2002**

**[6] Brzev et al., 2013 (GEM)**

**[7] Burton and Silva, 2014 (GEM)**

**[8] Colombi et al., 2008**

**[9] Damm, 2009**

**[10] De Leon and Carlos, 2006 (used by CAPRA)**

**[11] FEMA Earthquake model, 2013**

**[12] FEMA Flood model, 2013**

**[13] GEM, 2016**

**[14] Hahn, 2003 (used by CAPRA)**

**[15] Kircher et al., 1997**

**[16] Kircher et al., 2006 (HAZUS-MH)**

**[17] Lagomarsino et al., 2006**

**[18] Marulanda et al., 2013 (CAPRA)**

**[19] Menoni et al., 2002**

**[20] Merz et al., 2013**

**[21] Nastev and Todorov, 2013 (HAZUS-MH)**

**[22] Peng, 2012**

**[23] Penning-Rowsell et al., 2010**

**[24] Pitilakis et al., 2014**

**[25] Porter et al., 2008 (PAGER)**

**[26] Rashed and Weeks, 2003**

**[27] Scawthorn et al., 2006a (HAZUS-MH)**

**[28] Scawthorn et al., 2006b (HAZUS-MH)**

**[29] Spence et al., 2008 (GEVES)**

**[30] Yücemen et al., 2004**

**[31] See also Merz et al., 2010 for other selected reference**

| Vulnerability Indicator category | FLOOD VULNERABILITY | | EARTHQUAKE VULNERABILITY | |
|---|---|---|---|---|
| | Vulnerability curves | Index | Vulnerability curves | Index |
| *Demographics* | • Age (Agg, L) [12,26,27] <br><br> • # Vulnerable age (e.g. HAZUS: <16, >65) (Agg, L) [12, 26, 27] <br><br> • # Households (Agg, L) [12, 26, 27] <br><br> • Ethnicity (Agg, L) [12, 26, 27] | • Pre-existing health problems (Agg, L) [23] <br><br> • # Vulnerable age (e.g. MCM: > 75) (Agg, L-R) [9, 19,23, 29, 31] <br><br> • # Children (<14yr) (Agg, R) [19] <br><br> • # Elderly (>65yr) (Agg, R) [19] <br><br> • # Disabled (Agg, L) [3, 4, 31] <br><br> • Single parents (Agg, L) [23, 31] <br><br> • Household size (Agg, R) [19] <br><br> • % Pop. access sanitation (Agg, L) [2] <br><br> • Illiteracy rate (Agg, R) [1] <br><br> • Population density (Agg, R-G) [2, 29, 30] <br><br> • Size of urbanized area (Agg, R) [2, 29] <br><br> • % People in urban areas (Agg, R) [12] <br><br> • Ethnicity (Agg, L) [29] | • Age (Agg, L) [11, 17, 20] <br><br> • # People in vulnerable age range (e.g. HAZUS: <16, >65) (Agg, L) [11, 17, 20] <br><br> • # Households (Agg, L) [11, 17] <br><br> • Ethnicity (e.g. HAZUS) (Agg, L) [11, 17] <br><br> • Female population (Agg, L) [11, 17, 20] | • % Vulnerable age (e.g.< 5, >65) (Agg, L-N) [6, 9, 14, 16, 18, 28] <br><br> • % Households vulnerable age (Ob, L) [5] <br><br> • % Institutionalized elderly (Agg, L-R) [28] <br><br> • % Disabled (Agg, N) [6] <br><br> • # People per household/house (Agg, L-N) [6, 14, 16, 18, 28] (Ob,L) [5] <br><br> • Ethnicity (Agg, L-N) [6, 16, 28] <br><br> • % Immigrants (Agg, L-N) [6, 28] (Ob, L) [10] <br><br> • % Female (Agg, L-N) [6, 28] <br><br> • % Female headed household (Agg, L-N) [6, 28] <br><br> • % Population in poverty (Agg, L-R) [28] <br><br> • Access to education (Agg, L-N) [14] <br><br> • Education level (Agg, L-N) [6, 14, 16, 28] and (Ob, L) [5] <br><br> • Population density (Agg, L-N) [6, 7, 18, 22, 24] <br><br> • % Rural farm population (Agg, L-R) [28] <br><br> • % of Urban growth (Agg, N) [21] <br><br> • % Urban population (Agg, L-R) [28] <br><br> • Agricultural acreage (Agg, R) [21, 22, 25] <br><br> • % rural farm population (Agg, R) [32] |
| *Awareness* | | • Awareness and preparedness (Agg, L-R) [1, 3, 4, 20, 23] <br><br> • Access to information (phone/tv/radio) (Agg, L) [1, 2] | | • Emergency preparedness (Agg, L-R) [18] <br><br> • Access to information (last month's internet usage (Ob, L) [5] <br><br> • Household disaster-related |

| | | | | |
|---|---|---|---|---|
| | | • Past experience (Agg, L) [2, 20, 23]<br>• Pre-disaster coping strategies (Agg, L) [23]<br>• Existence of early warning systems (Agg, L-R) [4, 19, 34] | | attitudes, behaviours, customs and believes (Ob, L) [10]<br>• Ratio of expected financial loss to the total insured value (Agg, N) [31] |
| *Socio-economics* | • # Households per income classes (Agg, L) [12, 26, 27]<br>• # people working in commercial and industry (Agg, L) [12, 26, 27]<br>• % Rental / home owners (Agg, L) [12, 26, 27]<br>• Non-car ownership (Agg, L) [26, 27] | • Monthly net income (Agg, L-R) [2, 19, 29]<br>• % Unemployment (Agg, L) [23, 31]<br>• Housing ownership structure (Agg, L-R) [19, 31]<br>• Non-car ownership. (Agg, L) [31]<br>• Socioecon. status (Agg, R) [19]<br>• GDP (Agg, L-G) [2, 15] and (Agg, N) [13, 21]<br>• GINI coefficient (Agg, N) [13]<br>• Welfare level (Agg, R) [1]<br>• Percent with less than 12th grade education (Agg, L) [29]<br>• Centrality of an economic activity in a network (Agg, R) [33] | • # Households per income classes (Agg, L) [11, 17]<br>• # House rental / owners (Agg, L) [11,17]<br>• # grad. students (Agg, L) [11, 17]<br>• # students College (Agg, L) [11, 17]<br>• Sector-specific capital dependency (Agg, L-N) [14]<br>• Sector-specific labour dependency (Agg, L-N) [14]<br>• Sector-specific supply chain dependency (Agg, L-N) [14]<br>• Sector-specific infrastructure dependency (Agg, L-N) [14]<br>• # People in commercial and industry (Agg, L) [11, 17] | • Household wealth (e.g. private toilet) (Ob, L) [5]<br>• Income distribution (Agg, L-N) [9, 14, 28] and (Ob, L) [5, 10]<br>• % Unemployment (Agg, L-R) [6, 16, 28] and (Ob, L) [10]<br>• % Household social security (Agg, L-N) [6, 28]<br>• % Rental housing units (Agg, L-R) [6, 16, 28] and (Ob, L) [10]<br>• Median gross rent (US$) (Agg, L-R) [28]<br>• % Employed industry (farming, fishing, mining) (Agg, L-R) [28]<br>• % Employed secondary industry (Agg, N) [6]<br>• % Female labour force participation / unemployed (Agg, L-N) [6, 28]<br>• % People employed in transportation, communications, public utilities (Agg, L-R) [28] |
| *Institutional and political* | | • Urban planning institutions Y/N? (Agg, L) [2, 23]<br>• Investments in precautionary measures (Agg, L) [8] | | • Political stability (Agg, L-N) [14]<br>• Crime rate (Agg, N) [6] |

**Table 2: Overview of social earthquake and flood vulnerability assessment indicators.**

**Selected references:**

**[1] Akukwe and Ogbodo, 2015**

**[2] Balica et al., 2009**

**[3] Balica et al., 2012**

**[4] Barroca et al., 2008**

**[5] Brink and Davidson, 2015**

**[6] Burton and Silva, 2014 (GEM)**

**[7] Carreño, 2012**

**[8] Connor and Hiroki, 2005**

**[9] Davidson and Shah, 1997**

**[10] Duzgun et al., 2011**

**[11] FEMA Earthquake model, 2013**

**[12] FEMA Flood model, 2013**

**[13] Ferreira et al., 2011**

**[14] GEM, 2016**

[15] Jongman et al., 2015

[16] Khazai et al., 2014 (SYNER-G)

[17] Kircher et al., 2006 (HAZUS-MH)

[18] Menoni and Pergalani, 1996

[19] Merz et al., 2013

[20] Nastev and Todorov, 2013 (HAZUS-MH)

[21] Peduzzi, 2009 (GEM)

[22] Peng, 2012

[23] Penning-Rowsell et al., 2010

[24] Pergalani, 1996

[25] Rose et al., 1997

[26] Scawthorn et al., 2006a (HAZUS-MH)

[27] Scawthorn et al., 2006b (HAZUS-MH)

[28] Schmidtlein et al., 2011

[29] de Sherbinin and Bardy, 2015

[30] Spence et al., 2008 (GEVES)

[31] Tapsall, 2002

[32] Tierney and Nigg (1995)

[33] Van der Veen and Logtmeijer, 2005

[34] See also Merz et al., 2010 for other selected reference

| | Indicators at object scale | Physical indicators for buildings | Awareness related indicators | Social welfare and security indicators | Economic indicators | Time of the day | Changing exposure over time |
|---|---|---|---|---|---|---|---|
| **EQ** | + | + | - | - | + | + | - |
| **FL** | - | - | + | + | - | - | + |
| | **1. Flood vulnerability assessments could benefit from including more object scale indicators.** | **2. Flood vulnerability assessments could benefit from including more building-level indicators.** | **3. Earthquake vulnerability assessments could benefit from including more awareness related indicators.** | **4. Earthquake vulnerability assessments could benefit from including more social indicators** | **5. Flood vulnerability assessments could benefit from including more economic indicators.** | **6. Flood vulnerability assessments could benefit from including a time of the day indicator.** | **7. Earthquake vulnerability assessments could benefit from including a changing exposure over time indicator.** |

**Table 3: The + and – symbols depict the general occurrence of an indicator per hazard type as we concluded based on our literature review.**