# Peer review of "Review Article: A Comparison of Flood and Earthquake Vulnerability Assessment Indicators"

_Natural Hazards and Earth System Sciences, 2017_

## Referee Comment (RC1) · Anonymous Referee #1 · 5 Feb 2017

The review of vulnerability indicators in this paper is competent. The novelty of the contribution lies in the attempt to compare approaches for earthquakes and floods, and to see what lessons can be transferred from one to the other. This is quite valuable and moderately innovative, and the paper is generally well written, with a few minor lapses.

I tend to disagree with the fundamental basis of the approach adopted in this paper, in which vulnerability is broken down into sectors - physical, social, psychological, environmental, technical, environmental, etc. - and then recombined. I believe this is inefficient and it glosses over processes that involve several of the sectors at once. A better way to classify vulnerability is based on process (Alexander 1997, p. 292). For example, vulnerability can be seen in relation to the approach taken to manage it, or in

relation to factors that enhance it such as corruption, organised crime and technofixes. Another factor that is increasingly important is the cascading disaster. The principal vulnerability may lie at the escalation point, not in relation to the triggering event (Pescaroli and Alexander 2016). With the increasing complexity and interconnectedness of society, cascading disasters are going to become very important indeed.

As this is a review paper, the authors might consider examining a few references that have been left out (Cardona and Carreño 2011, Holand 2015, Kappes et al. 2011). Without wishing to suggest huge extensions, I feel uneasy about the lack of reference to the parallel development of resilience indicators. This is now a favourite topic of authors in the DRR field and, of course, it reflects the 'other side of the coin' with respect to vulnerability indicators.

Specific points:-

Lines 13-14: "Next, a selection of index- and curve based vulnerability models that use these indicators have been described" - has been described

Lines 29-47: There is confusion between hazard and vulnerability here. The wording needs to be sorted out. The authors should refer here to some of the work of Roger Pielke Jr on assessing trends in hazard and vulnerability.

Line 50: vulnerability curves, conceptualised in engineering as fragility curves

Lines 115-120: Indicators for cascading disasters and their escalation points are needed. Line 131: "The vulnerability of both infrastructure and buildings are influenced" - is influenced.

References

Alexander, D.E. 1997. The study of natural disasters, 1977-97: some reflections on a changing field of knowledge. Disasters 21(4): 284-305.

Cardona, O.D. and M.L. Carreño 2011. Updating the indicators of disaster risk and risk

management for the Americas. Journal of Integrated Disaster Risk Management 1(1): 1-21.

Holand, I.S. 2015. Lifeline issue in social vulnerability indexing: a review of indicators and discussion of indicator application. Natural Hazards Review 16(3): 1-12.

Kappes, M.S., M. Papathoma-Köhle and M. Keiler 2011. Assessing physical vulnerability for multi-hazards using an indicator-based methodology. Applied Geography 32(2): 577-590.

Pescaroli, G. and D. Alexander 2016. Critical infrastructure, panarchies and the vulnerability paths of cascading disasters. Natural Hazards 82(1): 175-192.

---

## Referee Comment (RC2) · Anonymous Referee #2 · 1 Mar 2017

The article tries to highlight insights how earthquake and flood vulnerability indicators can be improved. This is generally laudable, to improve both types of indicators by achieving more integration and learning by examples from each other.

Overall this article is a bit problematic. It is a little worrying that it reiterates certain limited visions of vulnerability indicators and formula, focusing mainly on physical and exposure aspects, especially in the beginning of the article. The literature used is quite narrow for certain fields such as local level studies or social, economic or institutional vulnerability (and resilience) and while the article claims to be a review, it is quite limited in scope and missing insights from similar review approaches.

Content The authors distinguish social vulnerability into four groups. It is questionable

to put economic indicators under social vulnerability. The examples and reasoning provided come too short and examples for instance for institutional indicators are not fully convincing.

Some chapters like these or 2.2.3 are so short that the impression remains that they could rather be skipped. Three lines about the aspect of scale under a heading are not sufficient, especially, the reference and thorough discussion and link to several indices discussed before, is lacking. Moreover, the function of chapter 2.2.3 is unclear, since in chapter 3 those aspects are discussed (again) in much more detail.

In terms of argumentation, the paper and logic of language is often hard to follow; certain contradictions seem to appear. For example, in lines 285 ff. There are rather unsupported claims that building codes have not been observed in flood vulnerability studies. What does this include? Building codes for earthquakes? Or specific design codes for physical stability against flooding? Do such standards exit? Which ones? And have they really not been analysed? But this is just an example of the argumentation style in this paper; claims made within one sentence and then not detailed anymore or supported merely by one source – in this case one of the authors of this paper and on earthquake not flood vulnerability. Some contradiction is also in this sentence with the following sentence "while for floods Nikolowski (2014) provides an overview" So is knowledge available or not, is a bit unclear.

Text from 285 to 315: well, the authors cited here (from the same institutions as the authors) use earthquake models also in flood studies. But this is not justifying the argument the authors make; that there would exist no flood vulnerability indicators that also analyse built environment or road infrastructure or else. In fact, there are even papers out by the same institution that specifically analyse road vulnerability, but are not mentioned here (Keller and Atzl 2014 International journal of disaster risk science) This again underscores the main impression that this article leaves; limited in scope and line of justification as based on own work of the authors and certain colleagues who have a strong focus only on certain aspects of risk or vulnerability. Their focus is

fine, but this paper tries to be a review paper and should be much more balanced and informed by the diversity of approaches that exist.

State-of-the art: what about other review papers on vulnerability indicators such as Tate, de Sherbinin, or on similar resilience indicators etc. what did they find? What do UNISDR processes at the moment on indicators search for, demand, have achieved? The SREX report of IPCC and similar documents by Cardona and others have substantially contributed a joint understanding of vulnerability indicators on all types of hazards, and earthquake and floods are amongst the most prominent.

Method: it is not clear, how the table cells are justified – it is decisions by the authors to fill these cells and quite many of those appear to be based rather on assumptions and feelings by the authors, what should be emphasised or placed into a box. Is this 'method' the right approach? Some of the authors are really strong in quantitative data analysis or case study approaches – wouldn't 'it be much more compelling to provide those arguments for better indicators based on real data or on cases?

A theoretical underpinning is lacking as well; the cited work by Bruenau et al 2003 might serve as a starting point or an analysis of conceptual frameworks who tried to structure vulnerability dimensions already and provide insights that physical and social and cultural and economic etc aspects must be combined in indictors. Davidsson and Shah 1997 are a classic; but many who tried to apply it have struggled with the application since physical and social and exposure and hazard are often overlapping; where are the existing lessons learned studies here? A section also about the pitfalls and advances made?

Scientific language and style of argumentation needs major improvement. Sentences such as in line 326 are an example: "However, building age does not appear to be an important vulnerability indicator used in flood vulnerability assessments." They do not "appear to be": how do they come to this conclusion? How exactly is this to be derived from the previous sentence?

Abstract: "In a cross-discipline study" please name the disciplines later on in detail and explain a bit how there might exist differences in focus.

Conclusion: I suggest a much more balanced differentiation and more caution. Sentences such as "Flood vulnerability assessments have generally used a higher scale of geographical aggregation compared to earthquake vulnerability assessments." are wrong, if they are generalised. A great number of household level flood vulnerability indicator studies exist as do aggregated indices at multi-national level. Overall, the paper runs the risk to be limited in scope to characterise vulnerability assessments per se as physical vulnerability assessments. Maybe it would help if the authors provide a better delineation of their scope – regarding content, ambition, and countries and disciplines covered

Minor comments: Line 54: Source is Davidson and Shah 1997 Line 380: Author is Rufat?

---

## Referee Comment (RC3) · Anonymous Referee #3 · 15 Mar 2017

General comments

This manuscript proposes a comparative review of the vulnerability indicators that have been recently used in flood and earthquake vulnerability assessments, while distinguishing physical and social vulnerability indicators. The approach is based on a literature review of recent studies or vulnerability models, and the manuscript discusses which types of indicators are used in flood or in earthquake studies, and whether some lessons are to be respectively gained from these two fields.

The intent of the authors to examine and harmonize the research outcomes of several disciplines (i.e. earthquake risk and flood risk, engineering community and socio-economic community) is a timely and welcome effort, which should be of high interest

for the audience of the NHESS journal.

However, this review lacks context, in the sense that the objective of the vulnerability assessment is not clearly specified: Is it for a risk or loss analysis? With the quantification of what type of impacts (direct or indirect, tangible or intangible, etc.)? Short term or long term risk? Is resilience taken into account? The various references and studies that have been selected to extract vulnerability indicators are mentioned in the tables without any information on their objectives and context. As a result, the conclusions of the review are undermined by this limitation, since – in its present form – it is not possible to exactly know why some vulnerability indicators have been taken into account or omitted by the various studies/models. Moreover, the paper concludes that some vulnerability indicators from earthquake analyses should be taken into account for flood analysis (and vice versa), whereas there is no proof or demonstration that such indicators would actually be relevant or useful for the subsequent risk analysis: this highlights once again the need to specify the aim and context of the so-called "vulnerability assessment".

Regarding the form, the paper would benefit from a better presentation of the review results. Section 3 quickly becomes a long list of repetitive sentences, detailing which vulnerability indicator or model is mostly used for flood and earthquake studies. Therefore it is difficult for the reader to get a synthetic view of strong tendencies, which should be obtained from an in-depth analysis instead of solely a description of the content of the two tables. Moreover, one may argue that the availability of more or less advanced vulnerability models for flood or earthquake studies has a strong influence on the type of vulnerability indicators that are required – and thus collected in the various studies.

Specific comments

1. l. 155-160: Maybe the education level should be mentioned here as a vulnerability indicator, since it is discusser later on (Section 3).

2. l. 210-213: The discussion on vulnerability curves for flood damage holds in three

lines, while earthquake vulnerability curves are described in one page. The authors should clarify this discrepancy and state whether flood vulnerability models are much scarcer than earthquake ones (and why).

3. p 6-7: There seems to be some confusion between vulnerability curves and fragility curves, which are not exactly the same mathematical object. Vulnerability curves are usually deterministic models that express a loss or damage rate with respect to a hazard parameter, while fragility curves are probabilistic models that provide the conditional probability of reach a given (discrete) damage state given a hazard parameter. The distinction between vulnerability indices and vulnerability curves is also debatable: for instance, the vulnerability assessment method by Giovinazzi et al. first generates a vulnerability index for the buildings, which is then used to generate a vulnerability curve.

4. l. 250-268: This sub-section (2.2.3) stands out from the rest of the section and is difficult to understand as it is (e.g. only two sentences to detail scaling issues). The authors should either remove it or ensure a better link with the previous sub-sections.

5. There is very little mention of the non-structural components or building contents as vulnerability indicators, even though they are usually responsible for most losses in the case of floods.

6. Table 2: There is no mention of the social indicators that have been identified in the SYNER-G project, for the development of shelter demand or healthcare demand models (no reference of this project in the table). See for instance Khazai et al. (2014).

7. English language style: the grammatical construction 'noun-based noun' is abused throughout the paper, especially without a '-' in many instances. A good example is the sentence at lines 564-565. I advise the authors to correct this in order to simplify some sentences and improve general readability.

Technical corrections

- l.199: "and" is repeated twice. - l.357: "SYNER-G" instead of "SYNERG-G" - l.408: "take more indicators" instead of "make more indicators". - l.494: "damage models" instead of "damage modes". - l.517: "is introduced by" instead of "is introduces by".

References

Khazai, B., Daniell, J. E., Düzgün, Ş., Kunz-Plapp, T., & Wenzel, F. (2014). Framework for systemic socio-economic vulnerability and loss assessment. In SYNER-G: Systemic Seismic Vulnerability and Risk Assessment of Complex Urban, Utility, Lifeline Systems and Critical Facilities (pp. 89-130). Springer Netherlands.

---

## Author Response (AR1)

**Review Article: A Comparison of Flood and Earthquake Vulnerability Assessment Indicators**

Marleen C. de Ruiter, Philip J. Ward, James E. Daniell, Jeroen C. J. H. Aerts
* * *
**General comments**

*We would like to thank the reviewers for their very valuable comments. We acknowledge the fact that we were not clear enough in defining the scope of our paper and in particular our usage of a narrow definition of vulnerability and the focus on single-hazard type risk assessment models. We recognize that this may have caused confusion and therefore we have made the following general changes:*

- *We included a more explicit explanation of the scope of our paper: to conduct a literature review comparing methods for quantitatively assessing vulnerability in flood and earthquake risk assessments within which we look at both physical and social vulnerability aspects.*
- *Therefore, we have increased the depth of our analyses by adding 22 citations to support our statements and to bring more balance in the physical and social aspects of vulnerability in risk models. We included references suggested by the reviewers, such as:*
    - *Alexander, D. (1997). The study of natural disasters, 1977–97: Some reflections on a changing field of knowledge. Disasters, 21(4), 284-304.*
    - *Tate, E. (2012). Social vulnerability indices: a comparative assessment using uncertainty and sensitivity analysis. Natural Hazards, 63(2), 325-347.*
    - *de Sherbinin, A., & Bardy, G. (2015). Social vulnerability to floods in two coastal megacities: New York City and Mumbai. Vienna Yearbook of Population Research, 131-165.*
    - *Cardona, O. D. (2004). The need for rethinking the concepts of vulnerability and risk from a holistic perspective: a necessary review and criticism for effective risk management. Mapping vulnerability: Disasters, development and people, 17.*
    - *Cardona, O. D., & Carreño, M. L. (2011). Updating the indicators of disaster risk and risk management for the Americas. IDRIM Journal, 1(1), 27-47.*
- *We have removed contradictory comments to this goal.*
* * *
**Reviewer #1**

The review of vulnerability indicators in this paper is competent. The novelty of the contribution lies in the attempt to compare approaches for earthquakes and floods, and to see what lessons can be transferred from one to the other. This is quite valuable and moderately innovative, and the paper is generally well written, with a few minor lapses.

- *We thank the reviewer for the positive feedback and thorough comments, and are pleased that they value the scientific relevance of our research. The reviewer provides several very useful comments/suggestions for revisions and we have addressed these in the revised manuscript, as per our responses to each comment below.*

I tend to disagree with the fundamental basis of the approach adopted in this paper, in which vulnerability is broken down into sectors - physical, social, psychological, environmental, technical, environmental, etc. - and then recombined. I believe this is inefficient and it glosses over processes that involve several of the sectors at once. A better way to classify vulnerability is based on process (Alexander 1997, p. 292). For example, vulnerability can be seen in relation to the approach taken to manage it, or in relation to factors that enhance it such as corruption, organised crime and technofixes.

- *We acknowledge the strong interactions that exist between the different components of vulnerability, however in comparing different vulnerability assessments, we believe singling out the separate components of vulnerability and its indicators is merely done to simplify the ability to compare the different indicators rather than to disregard the existing interactions. Based on the review comment, we have added the following text (including line number for reference):*

[130] Several studies have discussed the approach to, and potential pitfalls in, defining different indicator categories (e.g. Davidsson and Shah, 1997; Bruneau et al., 2003; Birkmann, 2007). Bruneau et al. (2003) suggest a framework for the quantitative assessment of seismic resilience consisting of the following four interrelated dimensions of community resilience for which there exist no single measure (note: their definition of resilience overlaps in part with the definition of vulnerability used in this paper): technical, organization, social, and economic. Davidsson and Shah (1997) acknowledge the necessity of the development of "an index of vulnerability". Their Earthquake Disaster Risk Index (EDRI), a composite index, allows for the inclusion of different factors of vulnerability (i.e. physical infrastructure, population, economy and social-political system) (Davidsson and Shah, 1997). Davidsson and Shah (1997) too, acknowledge that factors (or classes) of vulnerability are not distinct entities and that there are many interactions, overlaps and contradictions between indicators from the different classes.  While acknowledging the difficulties in categorizing  vulnerability, we classify vulnerability indicators, similar to many flood and earthquake vulnerability assessments, in two main classes: (a) physical indicators that pertain directly to characteristics of the exposed assets, namely infrastructure and lifelines (including transportation infrastructure, utility lifelines, and essential lifelines) and buildings (including structural elements, occupancy, and environment related factors); and (b) social indicators, which include here: demographics, awareness, socio-economics, and institutional factors (e.g. Mileti, 1999; Cutter et al., 2003; Adger, 2006; Messner and Meyer, 2006; Roberts et al., 2009; Balica et al., 2012).

- *We thank the reviewer for bringing these relevant papers to our attention. We have added a sentence acknowledging that the processes involved in measuring quantitative vulnerability has its shortcomings and is much more complex than assumed in this paper as this is outside the scope of our study. We therefore added the following:*

**[81]** Most of the risk models, however, make simple assumptions on quantifying vulnerability, and have largely refrained from considering (changing) vulnerability as a potential cause of the growing impacts of floods (Koks et al., 2015b; Mechler and Bouwer, 2014). Several key challenges with the quantification of vulnerability to flooding include: (1) difficulties in developing meaningful and quantifiable indicators of vulnerability; (2) a lack of available and accurate data to measure those indicators, and the fact that the required data are often only available at highly aggregated levels; and (3) a lack of empirical data on flood losses to relate losses (damage) to vulnerability (Birkmann 2006; Thieken et al., 2008; Notaro et al., 2014).

- *We included the following references:*
  - Alexander, D. (1997). The study of natural disasters, 1977–97: Some reflections on a changing field of knowledge. Disasters, 21(4), 284-304.
  - Pescaroli, G., & Alexander, D. (2016). Critical infrastructure, panarchies and the vulnerability paths of cascading disasters. Natural Hazards, 82(1), 175-192.

Another factor that is increasingly important is the cascading disaster. The principal vulnerability may lie at the escalation point, not in relation to the triggering event (Pescaroli and Alexander, 2016). With the increasing complexity and interconnectedness of society, cascading disasters are going to become very important indeed.

- *We acknowledge the emergence of the scientific field studying cascading disasters (Pescaroli and Alexander, 2016) and agree there is a strong relationship between vulnerability and the propagation of cascading disasters (Pescaroli and Alexander, 2015). However, due to the complex nature of addressing cascading disasters, our study focuses on assessing and comparing separate single-hazard assessments rather than cascading ones. To that extent, we have added a sentence narrowing our scope to exclude cascading disasters and we better explained that the research focuses on single events, while acknowledging the importance of increasing the understanding of cascading events. We added the following sentences:*

**[107]** We recognize that the study of cascading events is an important, emerging field as discussed extensively in Pescaroli and Alexander (2016), however our focus is on single events only.

**[761]** More studies are looking into cascading events. We recognize this as an emerging field, and believe this field will benefit from further comparative research, involving more models and methods.

As this is a review paper, the authors might consider examining a few references that have been left out (Cardona and Carreño 2011, Holand 2015, Papathoma 2011).

- *We thank the reviewer for these recommendations and we have included the following references:*
  - Cardona, O. D., & Carreño, M. L. (2011). Updating the indicators of disaster risk and risk management for the Americas. IDRiM Journal, 1(1), 27-47.

  ○ Holand, I. S. (2014). Lifeline issue in social vulnerability indexing: A review of indicators and discussion of indicator application. Natural Hazards Review, 16(3), 04014026.

  ○ Papathoma-Köhle, M., Kappes, M., Keiler, M., & Glade, T. (2011). Physical vulnerability assessment for alpine hazards: state of the art and future needs. Natural Hazards, 58(2), 645-680.

Without wishing to suggest huge extensions, I feel uneasy about the lack of reference to the parallel development of resilience indicators. This is now a favourite topic of authors in the DRR field and, of course, it reflects the 'other side of the coin' with respect to vulnerability indicators.

- *We agree that this is very important, and have therefore added the sections outlined below. However, we are cautious to open up a discussion regarding the differences between resilience, and susceptibility and how they relate to vulnerability. We now carefully explain our focus on susceptibility in the introduction and method sections, and we now clearly state which work with the definitions of vulnerability and susceptibility as defined by UNISDR as shown by adding the following paragraphs:*

**[64]** While acknowledging the studies that further subdivide vulnerability into resilience and susceptibility, or that consider resilience to be vulnerability's counterpart (e.g. Fuchs 2009), we asses vulnerability as it is defined by UNISDR (2009), but we do account for both physical and socio-economic indicators of vulnerability.

**[207]** The definition of social vulnerability is much debated (Birkmann 2007). Hinkel (2011) states that although the debate around the conceptualization of social vulnerability continues to exist, agreement seems to have been reached on social vulnerability being context-specific and place-based as defined by Cutter et al. (2003). In this paper, we therefore use the definition of social vulnerability as provided by Cutter et al. (2003), where social vulnerability consists of social inequalities (i.e. social factors that influence peoples' susceptibility) and place inequality (i.e. factors such as urbanization and economic vitality that impact the social vulnerability of a place).

**[219]** Two research communities have assessed social vulnerability quite extensively: the climate change adaptation (CCA) community and the disaster risk reduction (DRR) research community (Turner et al., 2003; Thomalla et al., 2006; Mercer, 2010; Dewan, 2013). Concepts from both communities have become increasingly intertwined, integrating concepts of resilience and adaptive- or coping-capacity (e.g. Turner et al., 2003; Deressa, Hassan and Ringler, 2008; Kienberger et al., 2009; Merz et al., 2010; Scheuer et al., 2011; Brink and Davidson, 2015). Birkmann et al., (2013) provide an extensive overview of vulnerability perspectives and discuss the framing of vulnerability by both the DRR and CCA communities. Since many risk assessment models use the concept of susceptibility in assessing vulnerability (Birkmann et al., 2013) and since this is in line with the UNISDR (2009) definition of vulnerability, we will exclude a focus on resilience as a separate concept.

Lines 13-14: "Next, a selection of index- and curve based vulnerability models that use these indicators have been described" - has been described

- *We thank the reviewer for pointing this out and have adjusted the sentence accordingly.*

Lines 29-47: There is confusion between hazard and vulnerability here. The wording needs to be sorted out. The authors should refer here to some of the work of Roger Pielke Jr on assessing trends in hazard and vulnerability.

- *We thank the reviewer for their suggestions and acknowledge that our wording was not phrased carefully enough. We have adjusted the mentioned section and added the suggested citation.*

**[43]** In this paper, we use the widely applied definition of vulnerability as provided by UNISDR (2009). The paper specifically does not aim to produce another definition of vulnerability and we gratefully acknowledge the broad literature on vulnerability and previous discussions of definitions and conceptualizations of vulnerability (e.g. Alexander 1997; Cardona 2004; Cutter et al., 2003; Adger, 2006; Barroca et al., 2006; Birkmann et al., 2007; Hinkel, 2011).

**[49]** Many studies have suggested that the observed increase in risk in recent decades is mainly due to the increase in exposure of assets and people in hazard prone areas, and an increase in wealth (Pielke Jr and Downton, 2000; Kron, 2005; UNISDR, 2011; IPCC, 2012; Doocy et al., 2013b; Blaikie et al., 2014; MunichRe, 2014; Visser et al., 2014; GFDRR, 2016). To date, most studies on flood risk have found little signal for increasing hazard in the last decades (e.g. Kundzewicz et al., 2014; Jongman et al., 2015). However, recent research suggests that this could be due to the fact these studies have not accounted for changes in vulnerability over time (e.g. Mechler and Bouwer, 2014; Jongman et al., 2015) and the impact of risk reduction policies on flood damage and societal flood vulnerability is not well understood (Pielke Jr and Downton, 2000). Indeed, the quantification of vulnerability in risk assessments is known to be extremely difficult, which is why most studies assume constant vulnerability over time. Line 50: vulnerability curves, conceptualised in engineering as fragility curves

- *We acknowledge that we did not carefully explain the difference between vulnerability curves and fragility curves, nor how we have included them in our study. Therefore, we have adjusted the following two paragraphs.*

**[89]** Compared to other natural hazards, the quantification of vulnerability is most detailed for earthquake risk assessment models although challenges remain (Douglas 2007; Roberts et al., 2009). Historically, the assessment of physical vulnerability (often referred to as 'fragility') is well-developed and recently attempts have also been made to improve the quantification of social vulnerability (Sauter and Shah, 1987; Tiedemann, 1991; Yücemen et al., 2004; Carreño et al., 2005; Douglas, 2007; Roberts et al., 2009).

**[316]** Unlike most other hazard type risk assessments, earthquake risk assessments traditionally use fragility curves as a vulnerability, or expected damage, measure, in which probabilistic damage to, for example, buildings is related to a hazard parameter such as ground shaking intensity (Douglas, 2007). In this study, we grouped fragility curve-based models with other curve-based models.

Lines 115-120: Indicators for cascading disasters and their escalation points are needed.

- *Please see our earlier comments regarding cascading events.*

Line 131: "The vulnerability of both infrastructure and buildings are influenced" - is influenced.

- *We thank the reviewer for pointing this out and have adjusted the sentence accordingly.*

**Reviewer #2**

The article tries to highlight insights how earthquake and flood vulnerability indicators can be improved. This is generally laudable, to improve both types of indicators by achieving more integration and learning by examples from each other.

- *We thank the reviewer for recognizing the benefits of our research.*

Overall this article is a bit problematic. It is a little worrying that it reiterates certain limited visions of vulnerability indicators and formula, focusing mainly on physical and exposure aspects, especially in the beginning of the article. The literature used is quite narrow for certain fields such as local level studies or social, economic or institutional vulnerability (and resilience) and while the article claims to be a review, it is quite limited in scope and missing insights from similar review approaches.

- *The reviewer makes a valid remark, and vulnerability is indeed a very broad topic, with a wealth of literature. We have therefore decided to revise the focus of our paper on providing insights into how vulnerability indicators (both physical and social) are used in quantitative flood- and earthquake risk assessment models. Furthermore, following the reviewer's suggestions, we have made adjustments to better explain the revised scope in the abstract, sections 1 (introduction) and 4 (conclusions). In these sections 1 and 4 we now better explain:*
  - *the selection of models and usage of indicators only from studies that quantify vulnerability, as the goal is to improve quantitative risk assessment models.*
  - *that due to challenges in quantifying qualitative indicators, most studies use indicators that are often physical as these are more easily quantifiable than, for example, psychological vulnerability indicators. As a result, there is a focus in earthquake research on indicators stemming from physical vulnerability assessments.*
  - *that the main flood vulnerability indicators are applied to case studies with a less detailed spatial scale than earthquake vulnerability assessments where the application of vulnerability indicators are applied at more detailed spatial scale. As such, this forces us to include multiple scales (from local to national level) in trying to obtain cross-discipline lessons.*

- *In section 1, we have now added a paragraph on the focus of the paper which reads:*

**[60]** There are two distinct paradigms in assessing vulnerability: the natural sciences and the social sciences (Roberts et al., 2009). The former considers the human system to be passive, while exposed elements have varying vulnerability to a hazard which can differ in magnitude and is considered to be an active agent. In the social sciences approach to assessing vulnerability, the focus is on the coping capacity and resilience of the human system (Roberts et al., 2009). While acknowledging the studies that further subdivide vulnerability into resilience and susceptibility, or that consider resilience to be vulnerability's counterpart (e.g. Fuchs 2009), we asses vulnerability as it is defined by UNISDR (2009), but we do account for both physical and socio-economic indicators of vulnerability.

**[101]** The main goal of this study is to conduct a literature review to provide insights into how vulnerability indicators (both physical and social) are used in quantitative flood- and earthquake risk assessment models by comparing two different methods for quantitatively assessing vulnerability in flood and earthquake risk assessment models (i.e. curve- and index-based vulnerability assessments). It therefore does not aim to provide a comprehensive overview of all vulnerability indicators in the

domain of floods or earthquakes. Instead, we analyze only those indicators that have been addressed in both modeling domains and systematically assess the differences in using those indicators in both flood vulnerability and earthquake risk models.

Content The authors distinguish social vulnerability into four groups. It is questionable to put economic indicators under social vulnerability. The examples and reasoning provided come too short and examples for instance for institutional indicators are not fully convincing.

- *This is indeed a 'grey area', and as the reviewer acknowledges, indictor-categories aren't as clear cut as suggested by e.g. Davidsson and Shah (1997). Therefore, in quantitative assessments, economic indicators are often lumped in or have otherwise overlap with social, or socioeconomic, indicators. These discrepancies therefore also end up in our review. We agree with the reviewer that we did not carefully explain this and therefore, to support this claim, we have included an explanation based on work by others. We addressed this more carefully in sections 2.1.2 (social vulnerability indicators) and 3.1.2 (results social vulnerability indicators) by acknowledging the overlaps as they exist in the studies we reviewed. We also added new examples supporting our choice of subdividing social indicators using economic and institutional indicators. The relevant sections now read:*

[126] Several studies have discussed the approach to, and potential pitfalls in, defining different indicator categories (e.g. Davidsson and Shah, 1997; Bruneau et al., 2003; Birkmann, 2007). Bruneau et al. (2003) suggest a framework for the quantitative assessment of seismic resilience consisting of the following four interrelated dimensions of community resilience for which there exist no single measure (note: their definition of resilience overlaps in part with the definition of vulnerability used in this paper): technical, organization, social, and economic. Davidsson and Shah (1997) acknowledge the necessity of the development of "an index of vulnerability". Their Earthquake Disaster Risk Index (EDRI), a composite index, allows for the inclusion of different factors of vulnerability (i.e. physical infrastructure, population, economy and social-political system) (Davidsson and Shah, 1997). Davidsson and Shah (1997) too, acknowledge that factors (or classes) of vulnerability are not distinct entities and that there are many interactions, overlaps and contradictions between indicators from the different classes. While acknowledging the difficulties in categorizing vulnerability, we classify vulnerability indicators, similar to many flood and earthquake vulnerability assessments, in two main classes: (a) physical indicators that pertain directly to characteristics of the exposed assets, namely infrastructure and lifelines (including transportation infrastructure, utility lifelines, and essential lifelines) and buildings (including structural elements, occupancy, and environment related factors); and (b) social indicators, which include here: demographics, awareness, socio-economics, and institutional factors (e.g. Mileti, 1999; Cutter et al., 2003; Adger, 2006; Messner and Meyer, 2006; Roberts et al., 2009; Balica et al., 2012).

[207] The definition of social vulnerability is much debated (Birkmann 2007). Hinkel (2011) states that although the debate around the conceptualization of social vulnerability continues to exist, agreement seems to have been reached on social vulnerability being context-specific and place-based as defined by Cutter et al. (2003). In this paper, we therefore use the definition of social vulnerability as provided by Cutter et al. (2003), where social vulnerability consists of social inequalities (i.e. social factors that influence peoples' susceptibility) and place inequality (i.e. factors such as urbanization and economic vitality that impact the social vulnerability of a place).

**[229]** Reviewing the existing studies, there is no consensus on which aspects to include in social vulnerability. Many studies incorporate different combinations of social indicators (such as vulnerable age groups, population density and population growth) with political, environmental and/or economic indicators (e.g. Davidsson and Shah, 1999; Cardona 2006; Peduzzi et al., 2009). Based on this, we here distinguish four main social vulnerability indicator groups: demographic, awareness and preparedness, socio-economic, and institutional and political vulnerability.

Some chapters like these or 2.2.3 are so short that the impression remains that they could rather be skipped. Three lines about the aspect of scale under a heading are not sufficient, especially, the reference and thorough discussion and link to several indices discussed before, is lacking. Moreover, the function of chapter 2.2.3 is unclear, since in chapter 3 those aspects are discussed (again) in much more detail.

- *We agree with the reviewer and this section has been removed while keeping the relevant text in section 3.2.2 which discusses spatial and temporal scales.*

In terms of argumentation, the paper and logic of language is often hard to follow; certain contradictions seem to appear. For example, in lines 285 ff. There are rather unsupported claims that building codes have not been observed in flood vulnerability studies. What does this include? Building codes for earthquakes? Or specific design codes for physical stability against flooding? Do such standards exit? Which ones? And have they really not been analysed? But this is just an example of the argumentation style in this paper; claims made within one sentence and then not detailed anymore or supported merely by one source – in this case one of the authors of this paper and on earthquake not flood vulnerability. Some contradiction is also in this sentence with the following sentence "while for floods Nikolowski (2014) provides an overview" So is knowledge available or not, is a bit unclear.

- *In cases of single referencing: additional references have been added to support claims made.*
- *In case of one-line arguments: arguments have been elaborated on and clarified.*
- *On the Nikolowski reference, we agree with the reviewer and we have adjusted the mentioned paragraph with the sentence containing the Nikolowski (2014) citation as follows:*

**[371]** Flood vulnerability assessments have seen a recent transition from focusing on traditional flood protection measures which aim to decrease the flood probability for an area to building-specific resilience measures (Ashley et al., 2007; Naumann et al., 2011). One example where this has been done is a study by Nikolowski (2014) which provides an overview of different ranges of building age and their flood vulnerability; structural (load carrying) and non-structural (mechanical) components; roof types; and building maintenance factors. For flood, vulnerability of building- or land-use types are often related to flood hazard indicators such as flood depth or flood velocity to estimate potential losses (e.g. Roos 2003; Barroca et al., 2006).

Text from 285 to 315: well, the authors cited here (from the same institutions as the authors) use earthquake models also in flood studies. But this is not justifying the argument the authors make; that there would exist no flood vulnerability indicators that also analyse built environment or road infrastructure or else. In fact, there are even papers out by the same institution that specifically analyse road vulnerability, but are not mentioned here (Keller and Atzl 2014 International journal of disaster risk science) This again underscores the main impression that this article leaves; limited in

scope and line of justification as based on own work of the authors and certain colleagues who have a strong focus only on certain aspects of risk or vulnerability. Their focus is fine, but this paper tries to be a review paper and should be much more balanced and informed by the diversity of approaches that exist.

- *We agree with the reviewer and rewrote the paragraph fine-tuning the claims made and included more references from other institutes than those related to the authors, among which the suggested citation as follows:*

**[391]** Infrastructure and lifeline indicators are used both in earthquake and flood vulnerability assessments, for example inHAZUS-MH. Atzl and Keller (2013) provide a framework which links social vulnerability to critical infrastructure and create indicators at the individual level for infrastructure-specific social vulnerability of commuters in Stuttgart (e.g. travel distance, availability of alternative transport, and number of available public transport lines). As shown in Table 1 and as argued in other work (Miletti, 1999), there are fewer flood vulnerability assessment studies including infrastructure related indicators compared to earthquake vulnerability assessments. Keller and Atzl (2014) add to the existing body of experimental research by assessing the causal relation between extreme precipitation events and the impacts on German infrastructure using an explanatory approach. In other studies, earthquake vulnerability assessment models are occasionally adopted in flood vulnerability models to address infrastructure risk (Merz et al., 2010). However, the knowledge gap continues to exist and there is a need for further research (Keller and Atzl (2014).

- *To assess the differences or similarities between earthquake and flood vulnerability models and the indicators used, we only include risk assessment models that include a vulnerability component consisting of physical and/or social indicators and that pertain to either of the two hazard types (or, such as HAZUS-MH, models that incorporate separate assessment models for different hazard types). We agree that we have not stated this clearly and have therefore addressed the scope in the abstract and in section 1 as follows:*

**[17]** In assessing the differences and similarities between indicators used in earthquake and flood vulnerability models, we only include models that separately assess either of the two hazard types.

**[101]** The main goal of this study is to conduct a literature review to provide insights into how vulnerability indicators (both physical and social) are used in quantitative flood- and earthquake risk assessment models by comparing two different methods for quantitatively assessing vulnerability in flood and earthquake risk assessment models (i.e. curve- and index-based vulnerability assessments). It therefore does not aim to provide a comprehensive overview of all vulnerability indicators in the domain of floods or earthquakes. Instead, we analyze only those indicators that have been addressed in both modeling domains and systematically assess the differences in using those indicators in both flood vulnerability and earthquake risk models. We recognize that the study of cascading events is an important, emerging field as discussed extensively in Pescaroli and Alexander (2016), however our focus is on single events only. More specifically, we analyze which vulnerability indicators have been addressed in such quantitative methods by comparing the fields of flood and earthquake risk assessment. Through this comparison, we hope that both fields can learn from each other's respective approaches, further developing vulnerability as an important component in risk modeling.

State-of-the art: what about other review papers on vulnerability indicators such as Tate, de Sherbinin, or on similar resilience indicators etc. what did they find? What do UNISDR processes at

the moment on indicators search for, demand, have achieved? The SREX report of IPCC and similar documents by Cardona and others have substantially contributed a joint understanding of vulnerability indicators on all types of hazards, and earthquake and floods are amongst the most prominent.

- *The suggested references as listed in the general comment to the reviewer have been added where appropriate as well to the indicator overview tables. For example:*

[40] A recent review of the Sendai framework by Mysiak et al. (2016) shows that one of the key components required, is to identify and increase understanding of the main vulnerability indicators that drive risk.

[459] Tate (2012) argues that the social vulnerability index is the social equivalent of the quantitative physical vulnerability assessment. In these indices, demographic data is often used to describe social, economic, political and institutional vulnerability. However, since there is a lack of systematic evaluation of how social vulnerability indices are constructed, little is known about how well these social vulnerability indices perform (Tate 2012). Tate (2012) concludes that most studies only provide limited justification for the inclusion of specific indicators. He argues that researchers should give more thought as to which social indicators to include as well as their statistical properties.

[467] To assess exposure differences to flooding and whether those who are most exposed also have the highest social vulnerability, de Sherbinin and Bardy (2015) apply their social vulnerability index using different sets of indicators to New York and Mumbai. Their method build on earlier work by Cutter et al. (2003) and the IPCC Special Report on Extreme Events Framework (IPCC 2012). Inclusion of indicators differed for the two cities and was often dependent on data availability and applicability to the case study (de Sherbinin and Bardy, 2015).

[592] An important aspect of vulnerability assessments is their spatial scale (Cutter et al., 1996). Vulnerability assessment models can be applied on different spatial scales (high versus low resolution) and using different data types (object versus aggregate, or raster, based). This is often dependent on data availability: particularly for social vulnerability indicators it is challenging to find high quality social vulnerability data for measuring those indicators at a local level (e.g. de Sherbinin and Brady, 2015).

Method: it is not clear, how the table cells are justified – it is decisions by the authors to fill these cells and quite many of those appear to be based rather on assumptions and feelings by the authors, what should be emphasised or placed into a box. Is this 'method' the right approach? Some of the authors are really strong in quantitative data analysis or case study approaches – wouldn't 'it be much more compelling to provide those arguments for better indicators based on real data or on cases?

A theoretical underpinning is lacking as well; the cited work by Bruenau et al 2003 might serve as a starting point or an analysis of conceptual frameworks who tried to structure vulnerability dimensions already and provide insights that physical and social and cultural and economic etc aspects must be combined in indictors. Davidsson and Shah 1997 are a classic; but many who tried to apply it have struggled with the application since physical and social and exposure and hazard are

often overlapping; where are the existing lessons learned studies here? A section also about the pitfalls and advances made?

- *In agreement with the reviewer, we elaborated on our scope setting, focusing on risk assessment models that have a vulnerability component where supported by the literature we distinguish two classifications: (1) physical versus social and (2) the sub-components vulnerability curves and indices. We also included theoretical underpinnings such as in the references provided by the reviewer to better explain and justify our revised scope. We also included a discussion on the difficulties of creating indicator categories without overlap. In restructuring our scope, we also added Bruneau et al. (2003) as suggested by the reviewer.*

[126] Several studies have discussed the approach to, and potential pitfalls in, defining different indicator categories (e.g. Davidsson and Shah, 1997; Bruneau et al., 2003; Birkmann, 2007). Bruneau et al. (2003) suggest a framework for the quantitative assessment of seismic resilience consisting of the following four interrelated dimensions of community resilience for which there exist no single measure (note: their definition of resilience overlaps in part with the definition of vulnerability used in this paper): technical, organization, social, and economic. Davidsson and Shah (1997) acknowledge the necessity of the development of "an index of vulnerability". Their Earthquake Disaster Risk Index (EDRI), a composite index, allows for the inclusion of different factors of vulnerability (i.e. physical infrastructure, population, economy and social-political system) (Davidsson and Shah, 1997). Davidsson and Shah (1997) too, acknowledge that factors (or classes) of vulnerability are not distinct entities and that there are many interactions, overlaps and contradictions between indicators from the different classes. While acknowledging the difficulties in categorizing vulnerability, we classify vulnerability indicators, similar to many flood and earthquake vulnerability assessments, in two main classes: (a) physical indicators that pertain directly to characteristics of the exposed assets, namely infrastructure and lifelines (including transportation infrastructure, utility lifelines, and essential lifelines) and buildings (including structural elements, occupancy, and environment related factors); and (b) social indicators, which include here: demographics, awareness, socio-economics, and institutional factors (e.g. Mileti, 1999; Cutter et al., 2003; Adger, 2006; Messner and Meyer, 2006; Roberts et al., 2009; Balica et al., 2012).

- *On the further justification of these two main categories, we have changed the categories in the table to better match the description in section 2.1.1 and elaborated on the method used for distinguishing the different classes in the table in that same section and in section 1. This has been explained more thoroughly by justifying choices pertaining both to the physical and social aspects of vulnerability in risk assessment. The following pieces of revised text underpin this revised description of categories:*

[89] Compared to other natural hazards, the quantification of vulnerability is most detailed for earthquake risk assessment models although challenges remain (Douglas 2007; Roberts et al., 2009). Historically, the assessment of physical vulnerability (often referred to as 'fragility') is well-developed and recently attempts have also been made to improve the quantification of social vulnerability (Sauter and Shah, 1987; Tiedemann, 1991; Yücemen et al., 2004; Carreño et al., 2005; Douglas, 2007; Roberts et al., 2009).

[162] Adger (1999) discusses how some indicators of vulnerability can also be both direct and indirect, such as social inequality, which can be a direct measure of the coping capacity of a household or community to respond to a disaster but it can also be interpreted as an indirect measure of increased

poverty and insecurity. Therefore, we have decided to omit the classification of indicators between direct and indirect as well as tangible versus intangible from this paper.

[169] The physical factor of vulnerability is the most thoroughly researched segment of vulnerability science, in part because physical vulnerability is more easily quantifiable than social vulnerability (Notaro et al., 2014), and relates to the physical vulnerability of the assets exposed to natural hazards – in our case floods and earthquakes. In accordance with several of the studies reviewed, we make a distinction in three main exposed assets: (a) infrastructure and lifelines; (b) buildings and their structural and occupancy components; and (c) environment (e.g. Davidson and Shah, 1997; Mileti 1999; Carreño et al., 2007; Douglas 2007).

[182] As mentioned, there are challenges in grouping indicators in distinct categories. Some studies perceive lifeline vulnerability as part of social vulnerability (e.g. Cutter et al., 2003; Holand 2014). For example, Holand (2014) defines lifeline vulnerability as the aspects of social vulnerability that are influenced by lifeline failure and he reviews common indicators used. He argues that there has been little discussion on how to measure lifeline vulnerability and distinguishes three lifeline indicator categories: (1) indicators addressing lifeline density and financial impacts caused by a natural disaster; (2) indicators measuring network redundancy and the potential for losing connectivity; and (3) indicators measuring travel time to facilities that provide critical services. Many of the studies reviewed by Holand (2014) group lifeline indicators with built environment or other physical indexes.

[229] Reviewing the existing studies, there is no consensus on which aspects to include in social vulnerability. Many studies incorporate different combinations of social indicators (such as vulnerable age groups, population density and population growth) with political, environmental and/or economic indicators (e.g. Davidsson and Shah, 1999; Cardona 2006; Peduzzi et al., 2009). Based on this, we here distinguish four main social vulnerability indicator groups: demographic, awareness and preparedness, socio-economic, and institutional and political vulnerability. However, as mentioned before, we recognize that indicator categories are not clear cut and overlaps continue to exist (Davidsson and Shah, 1997).

[276] It should be noted however, that in some studies an index is generated and subsequently incorporated in a vulnerability curve (e.g. Giovinazzi and Lagomarsino, 2004). In those cases, we classified the indicator used to construct the index in the index-based models category.

Scientific language and style of argumentation needs major improvement. Sentences such as in line 326 are an example: "However, building age does not appear to be an important vulnerability indicator used in flood vulnerability assessments." They do not "appear to be": how do they come to this conclusion? How exactly is this to be derived from the previous sentence?

- *We agree with the reviewer and have removed the sentence and included a more nuanced paragraph which reads as follows:*

[442] Within flood vulnerability assessments, some research have been conducted regarding non-structural damages and disaster risk reduction measures (e.g. building regulations pushing for flood-proofing) to reduce building content damages (Dawson et al., 2011). However, rather than using a separate indicator, several models include content damage by adjusting the shape of the damage curve or changing maximum damage values. HAZUS-MH uses a 0.5 factor for estimating residential content damages in relation to structural damages (Scawthorne et al., 2006) and this factor has also

been used by other studies (e.g. Penning-Rowsell et al., 2010; de Moel et al., 2014). The Damagescanner, a curve-based flood vulnerability assessment model, accounts for three types of flood-proofing measures (i.e. wet-proofing, dry proofing and a combination of the two) in assessing future potential for damages by adding damage reduction factors (0-1) (Poussin et al., 2012).

- *We also thoroughly checked the paper for one-line arguments and adjusted them accordingly.*

Abstract: "In a cross-discipline study" please name the disciplines later on in detail and explain a bit how there might exist differences in focus.

- *We have adjusted the abstract incorporating the reviewer's suggestions as follows:*

  **[10]** In a cross-disciplinary study, we carried out an extensive literature review to increase understanding of vulnerability indicators used in the disciplines of earthquake- and flood vulnerability assessments. We provide insights into potential improvements in both fields by identifying and comparing quantitative vulnerability indicators grouped into physical- and social categories. […]In assessing the differences and similarities between indicators used in earthquake and flood vulnerability models, we only include models that separately assess either of the two hazard types.

Conclusion: I suggest a much more balanced differentiation and more caution. Sentences such as "Flood vulnerability assessments have generally used a higher scale of geographical aggregation compared to earthquake vulnerability assessments." are wrong, if they are generalised. A great number of household level flood vulnerability indicator studies exist as do aggregated indices at multi-national level. Overall, the paper runs the risk to be limited in scope to characterise vulnerability assessments per se as physical vulnerability assessments. Maybe it would help if the authors provide a better delineation of their scope – regarding content, ambition, and countries and disciplines covered.

- *We have made large efforts to improve and better describe the scope of the revised paper; please see earlier comments for details.*
- *In agreement with the reviewer, we have adjusted the sentences mentioned in this comment have been addressed and we carried out a thorough read-through of the article.*
- *We aimed to include an equal number of physical as well as social studies and tried to have a balance between the number of earthquake and flood vulnerability models included despite some research suggesting that there are more earthquake risk assessment models than flood risk assessment models.*

Minor comments: Line 54: Source is Davidson and Shah 1997 Line 380: Author is Rufat?

- *We thank the reviewer for pointing this out and have adjusted the citations accordingly.*

**Reviewer #3**

This manuscript proposes a comparative review of the vulnerability indicators that have been recently used in flood and earthquake vulnerability assessments, while distinguishing physical and social vulnerability indicators. The approach is based on a literature review of recent studies or vulnerability models, and the manuscript discusses which types of indicators are used in flood or in earthquake studies, and whether some lessons are to be respectively gained from these two fields. The intent of the authors to examine and harmonize the research outcomes of several disciplines (i.e. earthquake risk and flood risk, engineering community and socioeconomic community) is a timely and welcome effort, which should be of high interest for the audience of the NHESS journal.

- *We thank the reviewer for the positive feedback and thorough comments, and are pleased that they value the scientific relevance of our research to the NHESS journal's audience. The reviewer provides several very useful comments/suggestions for revisions and we have addressed these in the revised manuscript, as per our responses to each comment below.*

However, this review lacks context, in the sense that the objective of the vulnerability assessment is not clearly specified: Is it for a risk or loss analysis? With the quantification of what type of impacts (direct or indirect, tangible or intangible, etc.)? Short term or long term risk? Is resilience taken into account? The various references and studies that have been selected to extract vulnerability indicators are mentioned in the tables without any information on their objectives and context. As a result, the conclusions of the review are undermined by this limitation, since – in its present form – it is not possible to exactly know why some vulnerability indicators have been taken into account or omitted by the various studies/models. Moreover, the paper concludes that some vulnerability indicators from earthquake analyses should be taken into account for flood analysis (and vice versa), whereas there is no proof or demonstration that such indicators would actually be relevant or useful for the subsequent risk analysis: this highlights once again the need to specify the aim and context of the so-called "vulnerability assessment".

- *We agree with the reviewer that we had not clearly stated our scope and objectives. Therefore, and in line with comments made by the other reviewers, we have elaborated on this. For example:*

**[64]** While acknowledging the studies that further subdivide vulnerability into resilience and susceptibility, or that consider resilience to be vulnerability's counterpart (e.g. Fuchs 2009), we asses vulnerability as it is defined by UNISDR (2009), but we do account for both physical and socio-economic indicators of vulnerability.

**[101]** The main goal of this study is to conduct a literature review to provide insights into how vulnerability indicators (both physical and social) are used in quantitative flood- and earthquake risk assessment models by comparing two different methods for quantitatively assessing vulnerability in flood and earthquake risk assessment models (i.e. curve- and index-based vulnerability assessments). It therefore does not aim to provide a comprehensive overview of all vulnerability indicators in the domain of floods or earthquakes. Instead, we analyze only those indicators that have been addressed in both modeling domains and systematically assess the differences in using those indicators in both flood vulnerability and earthquake risk models. We recognize that the study of cascading events is an important, emerging field as discussed extensively in Pescaroli and Alexander (2016), however our

focus is on single events only. More specifically, we analyze which vulnerability indicators have been addressed in such quantitative methods by comparing the fields of flood and earthquake risk assessment. Through this comparison, we hope that both fields can learn from each other's respective approaches, further developing vulnerability as an important component in risk modeling.

- *In agreement with the reviewer's comment, we added a section which discusses the four different impact types (direct, indirect, tangible and intangible) in more depth, as follows:*

**[144]** Vulnerability indicators can be categorized in direct versus indirect indicators. Where the engineering community has mainly addressed direct (or physical) damage, the economic research community has mainly addressed indirect (economic) damages (Koks et al., 2015a). In recent years, it has become more common for damage models to integrate both approaches (Koks et al., 2015a). [...] Adger (1999) discusses how some indicators of vulnerability can also be both direct and indirect, such as social inequality, which can be a direct measure of the coping capacity of a household or community to respond to a disaster but it can also be interpreted as an indirect measure of increased poverty and insecurity. Therefore, we have decided to omit the classification of indicators between direct and indirect as well as tangible versus intangible from this paper.

- *The reviewer is right, and we deliberately narrowed down our vulnerability research to exclude a focus on resilience as we are, as mentioned in our reply to the other two reviewers, cautious to open up a discussion regarding the differences between resilience, and susceptibility and how they relate to vulnerability. We now carefully explain our focus on susceptibility in the introduction and method sections, we focus on vulnerability as defined by UNISDR as pertaining to susceptibility.*

**[60]** There are two distinct paradigms in assessing vulnerability: the natural sciences and the social sciences (Roberts et al., 2009). The former considers the human system to be passive, while exposed elements have varying vulnerability to a hazard which can differ in magnitude and is considered to be an active agent. In the social sciences approach to assessing vulnerability, the focus is on the coping capacity and resilience of the human system (Roberts et al., 2009). While acknowledging the studies that further subdivide vulnerability into resilience and susceptibility, or that consider resilience to be vulnerability's counterpart (e.g. Fuchs 2009), we asses vulnerability as it is defined by UNISDR (2009), but we do account for both physical and socio-economic indicators of vulnerability.

**[64]** While acknowledging the studies that further subdivide vulnerability into resilience and susceptibility, or that consider resilience to be vulnerability's counterpart (e.g. Fuchs 2009), we asses vulnerability as it is defined by UNISDR (2009), but we do account for both physical and socio-economic indicators of vulnerability.

**[207]** The definition of social vulnerability is much debated (Birkmann 2007). Hinkel (2011) states that although the debate around the conceptualization of social vulnerability continues to exist, agreement seems to have been reached on social vulnerability being context-specific and place-based as defined by Cutter et al. (2003). In this paper, we therefore use the definition of social vulnerability as provided by Cutter et al. (2003), where social vulnerability consists of social inequalities (i.e. social factors that influence peoples' susceptibility) and place inequality (i.e. factors such as urbanization and economic vitality that impact the social vulnerability of a place).

**[224]** Birkmann et al., (2013) provide an extensive overview of vulnerability perspectives and discuss the framing of vulnerability by both the DRR and CCA communities. Since many risk assessment

models use the concept of susceptibility in assessing vulnerability (Birkmann et al., 2013) and since this is in line with the UNISDR (2009) definition of vulnerability, we will exclude a focus on resilience as a separate concept.

- *While recognizing the ambiguity in categorizing vulnerability indicators, we acknowledge that we didn't provide sufficient theoretical underpinning of the framework used in our analysis and applied to our tables. We have addressed this as follows:*

[revised manuscript text omitted]

Regarding the form, the paper would benefit from a better presentation of the review results. Section 3 quickly becomes a long list of repetitive sentences, detailing which vulnerability indicator or model is mostly used for flood and earthquake studies. Therefore it is difficult for the reader to get a synthetic view of strong tendencies, which should be obtained from an in-depth analysis instead of solely a description of the content of the two tables. Moreover, one may argue that the availability of more or less advanced vulnerability models for flood or earthquake studies has a strong influence on the type of vulnerability indicators that are required – and thus collected in the various studies.

- *As mentioned in our reply to reviewer two, we tried to have a balance between the number of earthquake and flood vulnerability models despite some research suggesting that there are more earthquake risk assessment models than flood risk assessment models. The tables, which we expanded on based on the reviewer's recommendations, attempt to create a comprehensive overview of the different indicators.*

  [270] Hollenstein (2005) reviewed vulnerability models for a wide range of natural hazards and found that there were far more earthquake vulnerability models (100+) than flood models (less than 20). We have aimed to include an equal number of earthquake and flood vulnerability models.

- *In adjusting section 2, by removing section 2.2.3 and by rewriting section 3 we hope to have improved the flow of the paper leading up to the results.*

Specific comments

1. l. 155-160: Maybe the education level should be mentioned here as a vulnerability indicator, since it is discusser later on (Section 3).

- *We agree that we could improve the flow by already mentioning education level in section two prior to discussing it in chapter 3. We therefore adjusted the paragraph which now reads:*

  [243] Research has shown that risk perception is an important factor for households to determine their level of preparation for natural hazard events (e.g. Balica et al., 2012; Bubeck et al., 2012). For example, the experience with previous events has a positive effect on the awareness level (Balica et al., 2009). In addition, access to information sources, such as TV, determines the knowledge and awareness of the hazard (e.g. Balica et al., 2009; Brink and Davidson, 2015). Education level was found to not only influence peoples' socio-economic vulnerability (e.g. Cutter et al., 2003) but also household awareness and preparedness levels (Rüstemli and Karanci, 1999; Shaw et al., 2004).

2. l. 210-213: The discussion on vulnerability curves for flood damage holds in three lines, while earthquake vulnerability curves are described in one page. The authors should clarify this discrepancy and state whether flood vulnerability models are much scarcer than earthquake ones (and why).

- *This was an oversight on our behalf and we adjusted the section by adding the following discussion of curve based flood vulnerability models:*

[334] There are many flood risk models that use vulnerability curves, such as Hazus-MH, the Multi-Coloured Manual (MCM), GLOFRIS, the Damagescanner and the European Flood Awareness System (EFAS) (Meyer and Messner, 2005; Jongman et al., 2012; Ward et al., 2013). The MCM by Penning-Rowsell et al. (2010) is the most advanced curve-based flood damage assessment method in Europe (Jongman et al., 2012). Similar to HAZUS-MH, the MCM is an object-based model where buildings are classified based on building usage (i.e. residential, commercial and industrial) (Meyer and Messner, 2005), however it uses absolute depth-damage curves to relate damage in British Pounds to water depth. The MCM does not include indirect flood damages but it does account for short and long flood durations (Meyer and Messner, 2005; Jongman et al., 2012).

3. p 6-7: There seems to be some confusion between vulnerability curves and fragility curves, which are not exactly the same mathematical object. Vulnerability curves are usually deterministic models that express a loss or damage rate with respect to a hazard parameter, while fragility curves are probabilistic models that provide the conditional probability of reach a given (discrete) damage state given a hazard parameter. The distinction between vulnerability indices and vulnerability curves is also debatable: for instance, the vulnerability assessment method by Giovinazzi et al. first generates a vulnerability index for the buildings, which is then used to generate a vulnerability curve.

- *We agree with the reviewer that we didn't clearly state the difference between vulnerability and fragility curves and how we incorporated the latter in our study. We adjusted the relevant section which now reads:*

[311] The vast majority of flood- and earthquake vulnerability assessment models are based on damage functions or fragility curves that relate the (mostly-) physical indicators described in Sect. 2.1 with hazard parameters (Douglas, 2007). In flood damage models, vulnerability is commonly calculated by relating flood depth to building or land-use type using vulnerability curves per exposed building- or land-use type. These curves provide estimates of potential damage. Occasionally, other hazard parameters such as velocity and duration are added (Merz et al., 2010; Jongman et al., 2012). Unlike most other hazard type risk assessments, earthquake risk assessments traditionally use fragility curves as a vulnerability, or expected damage, measure, in which probabilistic damage to, for example, buildings is related to a hazard parameter such as ground shaking intensity (Douglas, 2007). In this study, we grouped fragility curve-based models with other curve-based models.

- *We acknowledge that a proper explanation of how we deal with studies that combine vulnerability curves and indices was lacking. We therefore added the following which also incorporates the suggested reference:*

[277] It should be noted however, that in some studies an index is generated and subsequently incorporated in a vulnerability curve (e.g. Giovinazzi and Lagomarsino, 2004). In those cases, we classified the indicator used to construct the index in the index-based models category.

4. l. 250-268: This sub-section (2.2.3) stands out from the rest of the section and is difficult to understand as it is (e.g. only two sentences to detail scaling issues). The authors should either remove it or ensure a better link with the previous sub-sections.

- *We fully agree and in accordance with the comments of the other reviewers, we have removed this section.*

5. There is very little mention of the non-structural components or building contents as vulnerability indicators, even though they are usually responsible for most losses in the case of floods.

- The reviewer is right to point out that this was missing from our analysis and we have therefore incorporated a discussion of non-structural components as follows:

[442] Within flood vulnerability assessments, some research have been conducted regarding non-structural damages and disaster risk reduction measures (e.g. building regulations pushing for flood-proofing) to reduce building content damages (Dawson et al., 2011). However, rather than using a separate indicator, several models include content damage by adjusting the shape of the damage curve or changing maximum damage values. HAZUS-MH uses a 0.5 factor for estimating residential content damages in relation to structural damages (Scawthorne et al., 2006) and this factor has also been used by other studies (e.g. Penning-Rowsell et al., 2010; de Moel et al., 2014). The Damagescanner, a curve-based flood vulnerability assessment model, accounts for three types of flood-proofing measures (i.e. wet-proofing, dry proofing and a combination of the two) in assessing future potential for damages by adding damage reduction factors (0-1) (Poussin et al., 2012).

[371] Flood vulnerability assessments have seen a recent transition from focusing on traditional flood protection measures which aim to decrease the flood probability for an area to building-specific resilience measures (Ashley et al., 2007; Naumann et al., 2011). One example where this has been done is a study by Nikolowski (2014) which provides an overview of different ranges of building age and their flood vulnerability; structural (load carrying) and non-structural (mechanical) components; roof types; and building maintenance factors. For flood, vulnerability of building- or land-use types are often related to flood hazard indicators such as flood depth or flood velocity to estimate potential losses (e.g. Roos 2003; Barroca et al., 2006). 6. Table 2: There is no mention of the social indicators that have been identified in the SYNER-G project, for the development of shelter demand or healthcare demand models (no reference of this project in the table). See for instance Khazai et al. (2014).

- *We acknowledge this shortcoming and have addressed it by including the suggested reference, as follows:*

[532] Khazai et al. (2014) argue that for earthquakes, most often social vulnerability is integrated as a linear consequence function of physical damage (e.g. building damage causing casualties). For earthquake vulnerability, the index-based SYNER-G framework designed by Khazai et al. (2014) integrates physical and social indicators where both are assumed to be a direct function of hazard intensity, physical vulnerability and social vulnerability of the at risk population. For example, the expected number of post-disaster homeless people depends not only on the number of damaged buildings but also socio-economic indicators. Khazai et al. (2014) focus on including socio-economic indicators that can be quantified and harmonized at an EU-level and urban scale which led to the inclusion of more often used indicators such as household tenure (proportion of households living in self-owned or rented housing). Socio-economic indicators use aggregated data and are mostly used in index-based vulnerability assessments rather than in curve-based vulnerability assessments.

7. English language style: the grammatical construction 'noun-based noun' is abused throughout the paper, especially without a '-' in many instances. A good example is the sentence at lines 564-565. I

advise the authors to correct this in order to simplify some sentences and improve general readability.

- *We have read through the paper carefully and rephrased sentences that made use of that particular grammatical construction.*

Technical corrections

- l.199: "and" is repeated twice. - l.357: "SYNER-G" instead of "SYNERG-G" - l.408: "take more indicators" instead of "make more indicators". - l.494: "damage models" instead of "damage modes". - l.517: "is introduced by" instead of "is introduces by".

- *We thank the reviewer for pointing this out and have made the adjustments.*

[revised manuscript text omitted]

• Accessibility of utility lifeline (Ob, L) [19]
• Maintenance of utility lifeline (Ob, L) [19]
• Age of utility lifeline (Ob, L) [19]
• Closeness one utility to another (Ob, L) [19]
• # Lifelines on bridges and viaducts (Ob, L) [19]
• Accessibility of essential facilities (Ob, L) [19] |
| *Buildings* | • Building structural types (Ob, L) [12, 22, 23, 26, 27, 31]
• # of stories (Ob, L) [12, 23, 27, 28, 31]
• Building height (Ob, L) [12, 23, 27, 28, 31]
• Building age (Ob, L) [23]
• Foundation type (Ob, L) [12, 27, 28]
• Building occupancy (Ob, L) [12, 22, 23, 26, 27] | • Quality of building structure (Agg, L) [1, 20]
• # of stories (Agg, L) [20]
• Floor space of building (Agg, L) [20] Building occupancy class (Ob, L) [3, 4, 20] | • Building structural types (i.e. material) (Ob, L-G) [6, 8, 11, 13, 15, 16, 18, 21, 25]
• # of stories (Ob, L-G) [11, 13, 16, 18, 21]
• Building height (Ob, L-G) [6, 11, 13, 16, 18, 21]
• Building age (Ob, L) [10, 14]
• Roof type (Agg, G) [13] (Com, L-G) [10, 18]
• Building maintenance (Ob, L-R) [13, 17]
• Building configuration (Ob, L) [16, 21]
• Wall structural type (Com, L-G) [10, 18]
• Date of construction retrofit (Com, N) [6]
• Lateral load-resisting system (Com, N) [6]
• Building occupancy (Ob, L-G) [5, 6, 11, 16, 18, 29] | • # Stories (Agg, L-R) [30]
• # Stories above ground level (Agg, L-R) [30]
• Building height (Agg, L-R) [30]
• Roof type (Agg, L-R) [30]
• % of buildings in need of large repairs (Agg, N) [7]
• Soft story index (ratio of the ground story height to the first story height) (Agg, L-R) [30]
• Normalized redundancy score (Agg, L-R) [30]
• Min. norm. lateral stiffness index (Agg, L-R) [30]
• Overhang ratio (the floor area beyond outer frame / area ground fl (Agg, L-R) [30]
• Completed buildings in new constructions per 800 population (Agg, N) [7] |

| Environmental | | • Proximity to contaminating sites (Agg, R) [9]
• Types of vegetation (Agg, R) [2]
• Soil erosion potential (Agg, R) [9]
• Soil quality (Agg, R) [2, 4] | | • Proximity to contaminating sites (Ob, L) [19] |
| --- | --- | --- | --- | --- |

**Table 1: Overview of physical earthquake and flood vulnerability assessment indicators.**

[revised manuscript text omitted]

---

## Author Response (AR2)

Rebuttal round 2

**Review Article: A Comparison of Flood and Earthquake Vulnerability Assessment Indicators**

Marleen C. de Ruiter[1], Philip J. Ward[1], James E. Daniell[2], Jeroen C. J. H. Aerts[1]

[1]Institute for Environmental Studies (IVM), Vrije Universiteit Amsterdam, Amsterdam, 1081HV, The Netherlands
[2]Geophysical Institute and Center for Disaster Management and Risk Reduction Technology, Karlsruhe Institute of Technology (KIT), Karlsruhe, 76344, Germany

*Correspondence to:* Marleen C. de Ruiter (m.c.de.ruiter@vu.nl)

**Comments reviewer #2**

*The improvements made by the authors are very good and the paper has now progressed in content quality and depth, balanced decription and structure. The authors have taken great efforts and documented their improvements in an excellent manner. This paper is now really a good review and guidance document and should be published.*

*Very minor recommendations remain:*

*line 613: ‚floor' = flood?*

*line 717: „can easily be scaled up." How and in how far is this „easy"? Upscaling usually is difficult and has pitfalls.*

*Overall, the paper adresses a wide number of indicators in a rather brief way and keeping an overview is sometimes difficult. In order to add more depth and structure, another table, or, flow chart might help, showing how findings in the review of indicators lead to the recommendations in the end. Also, since some of the aspects in the recommendations have not been fully mentioned in the article before. However, given the efforts shown by the authors so far, this is optional.*

**Response**

We thank the reviewer for their valuable comments. We have made the adjustments accordingly and included a table to better explain how the seven conclusions followed from the findings of our literature review.

[revised manuscript text omitted]

 • Accessibility of utility lifeline (Ob, L) [19]
 • Maintenance of utility lifeline (Ob, L) [19]
 • Age of utility lifeline (Ob, L) [19]
 • Closeness one utility to another (Ob, L) [19]
 • # Lifelines on bridges and viaducts (Ob, L) [19]
 • Accessibility of essential facilities (Ob, L) [19] |
| *Buildings* | • Building structural types (Ob, L) [12, 22, 23, 26, 27, 31]
 • # of stories (Ob, L) [12, 23, 27, 28, 31]
 • Building height (Ob, L) [12, 23, 27, 28, 31]
 • Building age (Ob, L) [23]
 • Foundation type (Ob, L) [12, 27, 28]
 • Building occupancy (Ob, L) [12, 22, 23, 26, 27] | • Quality of building structure (Agg, L) [1, 20]
 • # of stories (Agg, L) [20]
 • Floor space of building (Agg, L) [20] Building occupancy class (Ob, L) [3, 4, 20] | • Building structural types (i.e. material) (Ob, L-G) [6, 8, 11, 13, 15, 16, 18, 21, 25]
 • # of stories (Ob, L-G) [11, 13, 16, 18, 21]
 • Building height (Ob, L-G) [6, 11, 13, 16, 18, 21]
 • Building age (Ob, L) [10, 14]
 • Roof type (Agg, G) [13] (Com, L-G) [10, 18]
 • Building maintenance (Ob, L-R) [13, 17]
 • Building configuration (Ob, L) [16, 21]
 • Wall structural type (Com, L-G) [10, 18]
 • Date of construction retrofit (Com, N) [6]
 • Lateral load-resisting system (Com, N) [6]
 • Building occupancy (Ob, L-G) [5, 6, 11, 16, 18, 29] | • # Stories (Agg, L-R) [30]
 • # Stories above ground level (Agg, L-R) [30]
 • Building height (Agg, L-R) [30]
 • Roof type (Agg, L-R) [30]
 • % of buildings in need of large repairs (Agg, N) [7]
 • Soft story index (ratio of the ground story height to the first story height) (Agg, L-R) [30]
 • Normalized redundancy score (Agg, L-R) [30]
 • Min. norm. lateral stiffness index (Agg, L-R) [30]
 • Overhang ratio (the floor area beyond outer frame / area ground fl (Agg, L-R) [30]
 • Completed buildings in new constructions per 800 population (Agg, N) [7] |

| Environmental | | • Proximity to contaminating sites (Agg, R) [9] | | • Proximity to contaminating sites (Ob, L) [19] |
|---|---|---|---|---|
| | | • Types of vegetation (Agg, R) [2] | | |
| | | • Soil erosion potential (Agg, R) [9] | | |
| | | • Soil quality (Agg, R) [2, 4] | | |

1235

**Table 1: Overview of physical earthquake and flood vulnerability assessment indicators.**

**Selected references:**

[1] **Akukwe and Ogbodo, 2015**

[2] **Balica et al., 2009**

[3] **Barroca et al., 2006 (FVAT)**

[4] **Barroca et al., 2008**

[5] **Bommer et al., 2002**

[6] **Brzev et al., 2013 (GEM)**

[7] **Burton and Silva, 2014 (GEM)**

[8] **Colombi et al., 2008**

[9] **Damm, 2009**

[10] **De Leon and Carlos, 2006 (used by CAPRA)**

[11] **FEMA Earthquake model, 2013**

[12] **FEMA Flood model, 2013**

[13] **GEM, 2016**

[14] **Hahn, 2003 (used by CAPRA)**

[15] **Kircher et al., 1997**

[16] **Kircher et al., 2006 (HAZUS-MH)**

[17] **Lagomarsino et al., 2006**

[18] **Marulanda et al., 2013 (CAPRA)**

[19] **Menoni et al., 2002**

[20] **Merz et al., 2013**

[21] **Nastev and Todorov, 2013 (HAZUS-MH)**

[22] **Peng, 2012**

[23] **Penning-Rowsell et al., 2010**

[24] **Pitilakis et al., 2014**

[25] **Porter et al., 2008 (PAGER)**

[26] **Rashed and Weeks, 2003**

[27] **Scawthorn et al., 2006a (HAZUS-MH)**

[28] **Scawthorn et al., 2006b (HAZUS-MH)**

[29] **Spence et al., 2008 (GEVES)**

[30] **Yücemen et al., 2004**

[31] **See also Merz et al., 2010 for other selected reference**

| Vulnerability Indicator category | FLOOD VULNERABILITY | | EARTHQUAKE VULNERABILITY | |
|---|---|---|---|---|
| | **Vulnerability curves** | **Index** | **Vulnerability curves** | **Index** |
| *Demographics* | • Age (Agg, L) [12,26,27]
 • # Vulnerable age (e.g. HAZUS: <16, >65) (Agg, L) [12, 26, 27]
 • # Households (Agg, L) [12, 26, 27]
 • Ethnicity (Agg, L) [12, 26, 27] | • Pre-existing health problems (Agg, L) [23]
 • # Vulnerable age (e.g. MCM: > 75) (Agg, L-R) [9, 19,23, 29, 31]
 • # Children (<14yr) (Agg, R) [19]
 • # Elderly (>65yr) (Agg, R) [19]
 • # Disabled (Agg, L) [3, 4, 31]
 • Single parents (Agg, L) [23, 31]
 • Household size (Agg, R) [19]
 • % Pop. access sanitation (Agg, L) [2]
 • Illiteracy rate (Agg, R) [1]
 • Population density (Agg, R-G) [2, 29, 30]
 • Size of urbanized area (Agg, R) [2, 29]
 • % People in urban areas (Agg, R) [12]
 • Ethnicity (Agg, L) [29] | • Age (Agg, L) [11, 17, 20]
 • # People in vulnerable age range (e.g. HAZUS: <16, >65) (Agg, L) [11, 17, 20]
 • # Households (Agg, L) [11, 17]
 • Ethnicity (e.g. HAZUS) (Agg, L) [11, 17]
 • Female population (Agg, L) [11, 17, 20] | • % Vulnerable age (e.g.< 5, >65) (Agg, L-N) [6, 9, 14, 16, 18, 28]
 • % Households vulnerable age (Ob, L) [5]
 • % Institutionalized elderly (Agg, L-R) [28]
 • % Disabled (Agg, N) [6]
 • # People per household/house (Agg, L-N) [6, 14, 16, 18, 28] (Ob,L) [5]
 • Ethnicity  (Agg, L-N) [6, 16, 28]
 • % Immigrants (Agg, L-N) [6, 28] (Ob, L) [10]
 • % Female (Agg, L-N) [6, 28]
 • % Female headed household (Agg, L-N) [6, 28]
 • % Population in poverty (Agg, L-R) [28]
 • Access to education (Agg, L-N) [14]
 • Education level (Agg, L-N) [6, 14, 16, 28] and (Ob, L) [5]
 • Population density (Agg, L-N) [6, 7, 18, 22, 24]
 • % Rural farm population (Agg, L-R) [28]
 • % of Urban growth (Agg, N) [21]
 • % Urban population (Agg, L-R) [28]
 • Agricultural acreage (Agg, R) [21, 22, 25]
 • % rural farm population (Agg, R) [32] |
| *Awareness* | | • Awareness and preparedness (Agg, L-R) [1, 3, 4, 20, 23]
 • Access to information (phone/tv/radio) (Agg, L) [1, 2] | | • Emergency preparedness (Agg, L-R) [18]
 • Access to information (last month's internet usage (Ob, L) [5]
 • Household disaster-related |

| | | | | |
|---|---|---|---|---|
| | | • Past experience (Agg, L) [2, 20, 23]
• Pre-disaster coping strategies (Agg, L) [23]
• Existence of early warning systems (Agg, L-R) [4, 19, 34] | | attitudes, behaviours, customs and believes (Ob, L) [10]
• Ratio of expected financial loss to the total insured value (Agg, N) [31] |
| *Socio-economics* | • # Households per income classes (Agg, L) [12, 26, 27]
• # people working in commercial and industry (Agg, L) [12, 26, 27]
• % Rental / home owners (Agg, L) [12, 26, 27]
• Non-car ownership (Agg, L) [26, 27] | • Monthly net income (Agg, L-R) [2, 19, 29]
• % Unemployment (Agg, L) [23, 31]
• Housing ownership structure (Agg, L-R) [19, 31]
• Non-car ownership. (Agg, L) [31]
• Socioecon. status (Agg, R) [19]
• GDP (Agg, L-G) [2, 15] and (Agg, N) [13, 21]
• GINI coefficient (Agg, N) [13]
• Welfare level (Agg, R) [1]
• Percent with less than 12th grade education (Agg, L) [29]
• Centrality of an economic activity in a network (Agg, R) [33] | • # Households per income classes (Agg, L) [11, 17]
• # House rental / owners (Agg, L) [11,17]
• # grad. students (Agg, L) [11, 17]
• # students College (Agg, L) [11, 17]
• Sector-specific capital dependency (Agg, L-N) [14]
• Sector-specific labour dependency (Agg, L-N) [14]
• Sector-specific supply chain dependency (Agg, L-N) [14]
• Sector-specific infrastructure dependency (Agg, L-N) [14]
• # People in commercial and industry (Agg, L) [11, 17] | • Household wealth (e.g. private toilet) (Ob, L) [5]
• Income distribution (Agg, L-N) [9, 14, 28] and (Ob, L) [5, 10]
• % Unemployment (Agg, L-R) [6, 16, 28] and (Ob, L) [10]
• % Household social security (Agg, L-N) [6, 28]
• % Rental housing units (Agg, L-R) [6, 16, 28] and (Ob, L) [10]
• Median gross rent (US$) (Agg, L-R) [28]
• % Employed industry (farming, fishing, mining) (Agg, L-R) [28]
• % Employed secondary industry (Agg, N) [6]
• % Female labour force participation / unemployed (Agg, L-N) [6, 28]
• % People employed in transportation, communications, public utilities (Agg, L-R) [28] |
| *Institutional and political* | | • Urban planning institutions Y/N? (Agg, L) [2, 23]
• Investments in precautionary measures (Agg, L) [8] | | • Political stability (Agg, L-N) [14]
• Crime rate (Agg, N) [6] |

**Table 2: Overview of social earthquake and flood vulnerability assessment indicators.**

**Selected references:**

| | | |
|---|---|---|
| [1] **Akukwe and Ogbodo, 2015** | [6] **Burton and Silva, 2014 (GEM)** | [11] **FEMA Earthquake model, 2013** |
| [2] **Balica et al., 2009** | [7] **Carreño, 2012** | |
| [3] **Balica et al., 2012** | [8] **Connor and Hiroki, 2005** | [12] **FEMA Flood model, 2013** |
| [4] **Barroca et al., 2008** | [9] **Davidson and Shah, 1997** | [13] **Ferreira et al., 2011** |
| [5] **Brink and Davidson, 2015** | [10] **Duzgun et al., 2011** | [14] **GEM, 2016** |

[15] Jongman et al., 2015

[16] Khazai et al., 2014 (SYNER-G)

[17] Kircher et al., 2006 (HAZUS-MH)

[18] Menoni and Pergalani, 1996

[19] Merz et al., 2013

[20] Nastev and Todorov, 2013 (HAZUS-MH)

[21] Peduzzi, 2009 (GEM)

[22] Peng, 2012

[23] Penning-Rowsell et al., 2010

[24] Pergalani, 1996

[25] Rose et al., 1997

[26] Scawthorn et al., 2006a (HAZUS-MH)

[27] Scawthorn et al., 2006b (HAZUS-MH)

[28] Schmidtlein et al., 2011

[29] de Sherbinin and Bardy, 2015

[30] Spence et al., 2008 (GEVES)

[31] Tapsall, 2002

[32] Tierney and Nigg (1995)

[33] Van der Veen and Logtmeijer, 2005

[34] See also Merz et al., 2010 for other selected references

|  | Indicators at object scale | Physical indicators for buildings | Awareness related indicators | Social welfare and security indicators | Economic indicators | Time of the day | Changing exposure over time |
|---|---|---|---|---|---|---|---|
| **EQ** | + | + | - | - | + | + | - |
| **FL** | - | - | + | + | - | - | + |
|  | **1. Flood vulnerability assessments could benefit from including more object scale indicators.** | **2. Flood vulnerability assessments could benefit from including more building-level indicators.** | **3. Earthquake vulnerability assessments could benefit from including more awareness related indicators.** | **4. Earthquake vulnerability assessments could benefit from including more social indicators** | **5. Flood vulnerability assessments could benefit from including more economic indicators.** | **6. Flood vulnerability assessments could benefit from including a time of the day indicator.** | **7. Earthquake vulnerability assessments could benefit from including a changing exposure over time indicator.** |

**Table 3: The + and – symbols depict the general occurrence of an indicator per hazard type as we concluded based on our literature review.**